# RESOURCE EFFICIENT SELF-SUPERVISED LEARNING FOR SPEECH RECOGNITION

## ABSTRACT

Representation learning from sequential data using self-supervised learning (SSL) has proven to be a powerful technique and improved state-of-the-art (SOTA) results when fine-tuned for various downstream tasks, including Automatic Speech Recognition (ASR). So far the success of SSL frameworks, e.g., Wav2Vec-2.0, for speech-to-text modeling is primarily carried out by masking intermediate features and then solving a contrastive task in an end-to-end manner. Although very successful, the overall training time (for example, days or weeks) and demanding resource requirements for achieving SOTA performance remain a significant barrier to further improving ASR solutions using such approaches. In this work we show that non-contrastive learning, such as an extension of the Barlow–Twins methodology, when applied to speech-to-text SSL modeling improves convergence, while reducing training time. Our results show that Wav2Vec-2.0 architecture pre-training with a non-contrastive SSL approach *reduces the GPU training hours by* 2.23 *times*, compared to masking based SSL approaches, while achieving a significant improvement (i.e., up to 6% relative WER decrease) in the model performance for the ASR task. We further demonstrate that a combination of both masking-based contrastive and non-contrastive SSL improves the ASR performance, e.g., up to 12% relative WER decrease, for all splits of LibriSpeech evaluation dataset.

## 1 INTRODUCTION

Modern industry-scale speech recognition systems often require tens-of-thousand hours of high quality labeled speech data to achieve acceptable deployment performance (Baevski et al., 2020; Ramos et al., 2022). However, large-scale data collection remains an extremely time consuming and costly procedure and does not scale as the number of languages to support increases. Furthermore, for a majority of the vast number of spoken languages, a large and high-quality training dataset is often unavailable (Babu et al., 2022). Thus, effective learning using primarily unlabeled data has been an important and long-standing research topic within the machine learning community, where the main emphasis is on learning good representations from unlabeled data and then fine-tuning using task dependent limited amount of labeled data.

Recent progress in self-supervised learning (SSL) has been highly successful in utilizing unlabeled data and demonstrated superior performance in the domains of computer vision (CV) (Chen et al., 2020; He et al., 2020; Chen & He, 2021), natural language processing (NLP) (Devlin et al., 2019; Lewis et al., 2019), and speech recognition (SR) (Liu et al., 2020a; Chung et al., 2021; Baevski et al., 2022; Schneider et al., 2019; Baevski et al., 2020). In particular, SSL-based approaches exploit abundance of unlabeled data to learn underlying representations, while using both *contrastive* and *non-contrastive* approaches (Jaiswal et al., 2020; Balestriero & LeCun, 2022). Especially, in the domain of ASR, masking based contrastive methods have emerged as the leading SSL approach and yielding current state-of-the-art solutions, e.g., Wav2Vec-2.0 (Baevski et al., 2020), and Hu-BERT (Hsu et al., 2021). The success of these approaches is mainly due to easy availability of large curated unlabeled open source datasets (Kearns, 2014; Panayotov et al., 2015; Kahn et al., 2020; Wang et al., 2021; Ardila et al., 2020), availability of industry-scale GPU infrastructures, improvements in the data training pipeline and scaling (e.g., data-, pipeline-, model-parallelism) of deep learning frameworks. However, the overall training time for achieving SOTA performance remains a significant barrier to further improving ASR solutions using contrastive SSL.

In this paper we present a technique for decreasing the training time of SSL based ASR systems by using a non-contrastive SSL method, rather than a contrastive method, for speech representation

learning. In general, non-contrastive SSL is heavily under represented in audio research, with a few notable exceptions (Liu et al., 2022). This is primarily due to the fact that SSL methods are more established in the domain of CV than audio applications and often require novel extensions. Towards bridging this gap, we consider Barlow–Twins (Zbontar et al., 2021) as a representative example of non-contrastive SSL and expand its scope from vision to audio by inventing the following extensions: (i) we incorporate a number of new loss functions via purposefully designed time-merging and time-unrolling, and (ii) applying static (hyper-parameter optimization) and dynamic (stop gradient) methodologies to balance the different scales in individual losses. We further explore the effect of sequential use of non-contrastive and contrastive training and observe improved performance, i.e., decreased word error rate (WER) when compared to solely non-contrastive or contrastive training, which is in line with recent work on SSL based speech representation learning for speaker verification rather than ASR (Zhang & Yu, 2022).

A summary of our main findings regarding the benefits of our non-contrastive SSL approach for speech representation learning are as follows: (i) non-contrastive SSL ASR yields a 2.23x training speed up compared with contrastive SSL ASR in our experiments, while simultaneously improving up to 6% relative WER (c.f., first and second rows of Tables 1 and 6), (ii), fewer required GPUs and smaller batch size reducing memory requirements in non-contrastive as opposed to contrastive methods (c.f., Table 3), and (iii) lowest WER achieved by sequentially combined approach followed by non-contrastive training (c.f., third rows of Tables 1, 6).

## 2 APPROACH

The most common SSL methods in speech considered are masking-based contrastive learning and autoregressive prediction based learning. In this work, we explore the potential of a non-contrastive SSL method for learning speech representations and its effectiveness on the downstream ASR.

### 2.1 MOTIVATION

Recent work in the area of non-contrastive SSL (e.g., BYOL (Grill et al., 2020), SimSiam (Chen & He, 2021), Barlow–Twins (Zbontar et al., 2021), DINO (Caron et al., 2021)) have shown remarkable capacity to learn powerful representations from only positive pairs, i.e., two augmented views of the same data point. Unlike contrastive SSL approaches that use negative pairs to prevent representational collapse, non-contrastive SSL approaches employ a dual pair of Siamese networks to process two augmented views of a data point and minimize their representational differences.

In general, contrastive SSL methods require large batch sizes, e.g. SimCLR (Chen et al., 2020) and MoCo (He et al., 2020), to achieve good performance. On the contrary, non-contrastive SSL approaches are comparatively more efficient and easy to train with smaller batches and reduced memory. As shown in (Zbontar et al., 2021), a non-contrastive SSL method such as Barlow–Twins could learn effectively with up to 16x smaller batch size.

### 2.2 METHOD

In this subsection, we present an overview of our approach for learning speech embeddings with a first of its kind non-contrastive SSL method designed for time series speech modeling (c.f., Figure 1 (b)) and its comparison with a standard non-contrastive SSL method for non-time series data such as images (see Figure 1 (a)).

Similar to all non-contrastive SSL methods used in vision, our approach for learning speech embeddings has a dual pair of Siamese networks referred to as online ($O$) and target ($T$) networks. Only the online network is trained via gradient descent and the target network employs a momentum encoder (He et al., 2020) that slowly follows the online network in a delayed fashion through an exponential moving average (EMA). The outputs of the online and target networks are then encouraged to learn good representations via a self-supervised loss function.

However, there are two key differences in our approach for learning speech embeddings compared to image embeddings, which can be categorized as modeling and learning differences. These differences are summarized below.

First, instead of performing augmentations in the input space (c.f. Figure 1 (a)), our solution operates in a latent space (see Figure 1 (b)). Specifically, we apply augmentation not directly on the

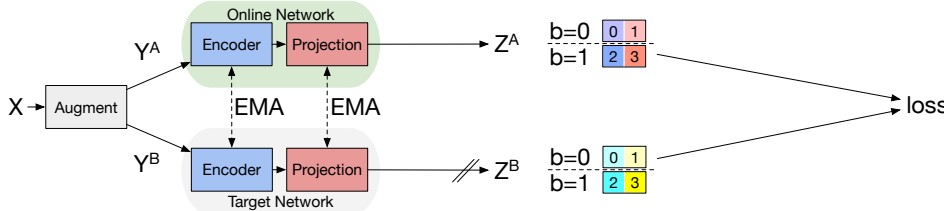

(a) Standard non-contrastive SSL approach with optional exponential moving average (EMA) and stop gradient (sg) to learn representation of non-time series data. Such an approach takes image data $X$ (depicted with batch size b=2), augments it into two distorted views $Y^A$ and $Y^B$, and feeds it via online and target networks to produce representational embeddings $Z^A$ and $Z^B$. An appropriate loss function is then created from these embeddings for self-supervised learning; particularly relevant to our work, the existing Barlow–Twins loss function cross-correlates $Z^A$ and $Z^B$ and promotes the correlation matrix to be close to identity.

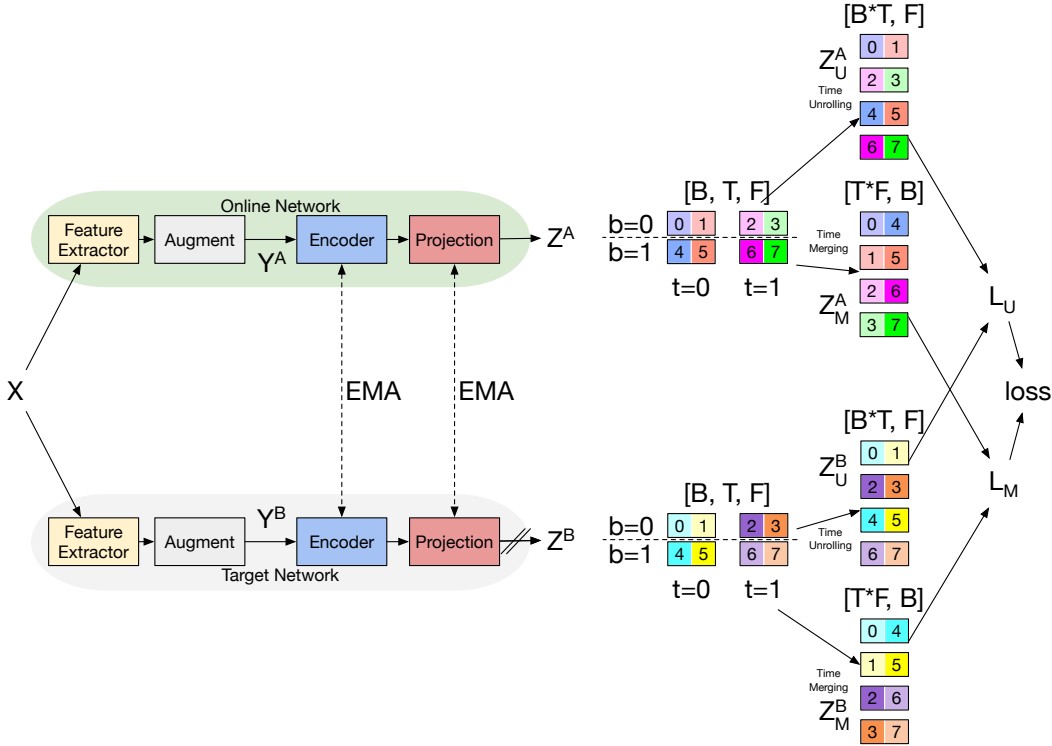

(b) Our approach for learning time series speech representations via non-contrastive SSL method. In the audio domain, a larger encoder is often composed of two submodules namely a feature extractor and a smaller encoder as depicted in the figure, and data augmentation employed in between. This motivates the changes to the online and target networks in our approach. Furthermore, traditional loss functions such as Barlow–Twins cannot be readily used given the dimensionality of the embeddings $Z$ is larger in the audio than the image domain. For this reason we introduce time unrolling and merging approaches leading each to a different loss function which we statically or dynamically combine for improved training.

Figure 1: Comparison of non-contrastive SSL approaches for (a) image and (b) audio data.

input $X$, but rather to the outputs of the feature extractor to generate $Y^A$ and $Y^B$. The motivation behind this is that the most common form of data augmentation in audio is SpecAugment (Park et al., 2019) which acts on the audio spectogram. However in the Wav2Vec-2.0 (Baevski et al., 2020) architecture the spectogram is replaced by a learnable sub-network referred to as the *feature extractor*. Nevertheless we have found that the same principles underlying SpecAugment are used at this point in the processing pipeline in order to avoid overfitting and increase generalization.

The second key difference stems from the way the loss is computed from the produced embeddings $Z^A$ and $Z^B$. Depending on if the scenario is non-time series or time series there is a fundamental difference between the tensor representation $Z^A$ and $Z^B$. Specifically, as shown in Figure 1 (a) $Z^A$ and $Z^B$ are two dimensional tensors, whereas in Figure 1 (b) $Z^A$ and $Z^B$ are three dimensional tensors due to an extra time dimension.

In order to address this difference, in this work we extend the application of Barlow–Twins loss for speech-to-text SSL modeling, in particular for speech (and more generally time varying) representation learning. As demonstrated in (Zbontar et al., 2021), for non time varying signals, one computes the cross-correlation matrix between the embeddings of online and target networks, and then optimizes the model such that this cross-correlation matrix comes close to the identity. For learning speech embeddings, we appropriately generalize the original Barlow–Twins loss by computing it from different views of the embedding data, namely as *time unrolling* and *time merging* approaches as depicted in Figure 1 (b), as well as in the text below.

i) **Time Unrolling.** In this approach, we unroll the sequence on time axis and stack each frame together. Specifically, a batch of outputs with dimensions $[B, T, F]$ is transformed to $[B * T, F]$. This transformation is performed for the outputs of both the online and target models. A cross correlation matrix $C^U$ of size $F \times F$ is then created and encouraged to be close to the identity via a loss function $L_U$. This enables the model to enforce diversity in each sequence, such that all frames cannot be the same. In other words, the time unrolling loss on the cross correlation matrix based on the reshaped tensor $[B * T, F]$ promotes that the $i - th$ feature across all the $B$ samples and $T$ timesteps is different from the $j - th$ feature.

ii) **Time Merging.** We merge together the features of each sequence on the time axis, so that the output of each utterance becomes one long tensor of features. We then compute a transpose of this as the Barlow–Twins loss is computed on the second dimension. Specifically, a batch of outputs with dimensions $[B, T, F]$ is transformed to $[T * F, B]$. Finally, we create a $B \times B$ correlation matrix $C^M$ and feed it to the loss function $L_M$. Through this loss we want the model to learn inter-sequence diversity such that the $i - th$ sample is different from the $j - th$ sample within the batch. This ensures that all sequences are not the same and thus SSL training does not collapse.

Formally, the loss function ($L$) can be defined as follows:

$$L_U \leftarrow \sum_i \frac{(1 - C_{ii}^U)^2}{N_U} + \sum_i \sum_{i \neq j} \frac{2(C_{ij}^U)^2}{N_U(N_U - 1)}, \text{ where } C^U \leftarrow \text{CrossCorrelation}(Z_U^A, Z_U^B)$$

$$L_M \leftarrow \sum_i \frac{(1 - C_{ii}^M)^2}{N_M} + \sum_i \sum_{i \neq j} \frac{2(C_{ij}^M)^2}{N_M(N_M - 1)}, \text{ where } C^M \leftarrow \text{CrossCorrelation}(Z_M^A, Z_M^B)$$

$$\text{loss} \leftarrow \begin{cases} w_U \cdot L_U + w_M \cdot L_M & \Leftarrow \text{static} \\ \frac{L_U}{\text{sg}(L_U)} + \frac{L_M}{\text{sg}(L_M)} & \Leftarrow \text{dynamic} \end{cases}$$

Here, $N_U$ and $N_M$ denote the first dimension of the square matrices $C^U$ and $C^M$ respectively. In practice we observe that the unrolling (U) loss $L_U$ and the merging (M) loss $L_M$ often have different scales, and for this reason we explore combining these loss functions statically or dynamically into an overall loss $L$. More specifically, for static scaling we simply multiply each loss function with a hyper-parameter weight $w_U$ for $L_U$ and $w_M$ for $L_M$ on which we conduct hyper-parameter optimisations (HPO). For dynamic scaling, we use the stop gradient (sg) operation to ensure each partial loss is divided by stop gradient applied to itself. Given that a scaling of the loss propagates to the size of the gradient update, this leads to overall better gradient update behavior without the need for HPO on the weight of each loss.

## 3 EVALUATION SETTINGS

### 3.1 ONLINE AND TARGET MODELS

We create the online and target networks via surgery on the Wav2Vec-2.0 Base architecture (Baevski et al., 2020). This is accomplished by taking the feature extractor and the encoder components of Wav2Vec-2.0 as well as additional augment and projection layers, which we organize in a sequential manner from feature extractor, augment layer, encoder and projection layer as described below.

**Feature Extractor.**   The feature extractor consists of seven convolution blocks. Each block has a temporal convolution layer followed by layer normalization and a GELU activation function (Hendrycks & Gimpel, 2016). The number of channels, strides and kernel widths are the same as in the Wav2Vec-2.0 architecture (Baevski et al., 2020). This results in the feature extractor's hop length of $20ms$ and the receptive field of $25ms$ of audio.

**Augmentation in Latent Space.**   An augmentation block is used to generate and apply masks on the feature extractor outputs. We use a masking function similar to SpecAugment (Park et al., 2019) and mask time-steps during training which avoid model collapse. We use the probability of $0.05$ and mask length of $10$ for the target model, and we use $0.1$ and $20$ for the online model. We use more masking for online network to bring highly distorted view close to its corresponding less distorted view, which is inspired from the local/global crop strategy presented in (Caron et al., 2021) and weak/strong augmentation strategy used in (Sohn et al., 2020). Note that, in order to avoids boundary effects, we apply augmentation only to the outputs of the feature extractor that mimics FFT style transformation with a receptive field of 25ms of audio and a 20ms hop length.

**Encoder.** We use the same encoder architecture as Wav2Vec-2.0 Base model which consists of 12 transformer blocks.

**Projection.** We project the output to a lower dimensional latent space. Specifically, we use a dense layer to transform the output of the encoder to a 29 dimensional space. This helps us to efficiently compute the loss on smaller output dimensions. Here, we selected 29 dimensional output as it is also the same dimension of final vocabulary (discussed later in 3.4).

## 3.2   DATASET

During the SSL phase, we pre-train the models using the LibriSpeech corpus (Panayotov et al., 2015) of 960 hours of audio (LS-960) without the transcriptions. We crop the audio utterances into 5 seconds of speech and batch them together. We do not perform any pre-processing since the feature extractor directly processes the raw audio data.

When fine-tuning the models, we consider 3 labeled data settings:

  i) LS-960: 960 hours of LibriSpeech data with transcriptions (Panayotov et al., 2015),

 ii) LS-100: the train-clean-100 subset of LibriSpeech comprising 100 hours of labeled data,

iii) LL-10: Libri-light limited resource training subset of 10 hours labeled dataset, which was originally extracted from LibriSpeech (Kahn et al., 2020).

We evaluate models on the standard LibriSpeech dev-clean/other and test-clean/other sets.

## 3.3   PRE-TRAINING

For pre-training the representation learning model, that includes the feature extractor and encoder networks, we consider three settings as described below.

  i) **W2V-2**: for this setting, we use the pre-trained weights of Wav2Vec-2.0 model, which is trained with masking based contrastive SSL method (Baevski et al., 2020). We use the checkpoint released in (Baevski et al., 2020) for Wav2Vec-2.0 Base model which was pre-trained on 960 hours of LibriSpeech corpus.

 ii) **Non-Contrastive**: this setting is based on our proposed non-contrastive SSL method for speech representation learning as described in Section 2.2.

iii) **Sequentially Combined**: in this setting, we consider pre-training the model with the *W2V-2* approach followed by our proposed *Non-Contrastive* setting. Specifically, we start with pre-trained weights from Wav2Vec-2.0 Base model (as described for *W2V-2* pre-training setting) and further pre-train with non-contrastive SSL method (i.e., *Non-Contrastive* setting). Note that there could be multiple ways to combine the contrastive and non-contrastive training such as non-contrastive followed by contrastive, vice versa, or in an iterative sequence with each contrastive or non-contrastive being applied a fixed number of steps or perhaps with a scheduler to switch between the two approaches in a more elaborate fashion. However, we leave these for further exploration as they are outside of the scope of the present work.

**Optimization.**   We use the ADAM optimizer (Kingma & Ba, 2015) with a learning rate of $10^{-5}$. The learning rate was selected between $10^{-5}$ and $5 \cdot 10^{-5}$ after training the models for $10K$ steps.

We use exponential-decay with decay rate of $0.99$ as a learning rate scheduler. For *Non-Contrastive* and *Sequentially Combined* approaches, we train the models for $400K$ and $250K$ steps. For both of these methods we use the batch size corresponding to 16 minutes of audio (details are described in Table 4). We observed that the convergence with dynamic scaling of the loss (discussed in 2.2) was better than the static option, which motivated us to use it throughout our experiments. We note however that it is possible that the static option could work with other values of $w_M$ and $w_U$ than the ones we tried. Specifically we experimented with nine combinations with $w_M$ and $w_U$ taking values in [0.1, 0.5, 1.0], so a more extensive HPO could perhaps improve on the dynamic scaling.

### 3.4 FINE-TUNING

For the fine-tuning phase, we take the target network without the projection layer and add a randomly initialized linear projection on top. This layer's outputs have the same dimensionality $V$ as the vocabulary of the downstream task. Specifically, for the ASR task we select $V$ as 29, which includes 26 characters A–Z, the white space, the apostrophe ($'$) as well as a blank symbol required for the connectionist temporal classification (CTC) described below.

We use CTC loss (Graves et al., 2006) and train the network by minimizing it. We keep the feature extractor block frozen during the fine-tuning and train the parameters of only the encoder and projection layer parts of the network (c.f., Figure 1 (b)). During the fine-tuning phase we keep the augmentation parameters the same as for the target model. However, during fine-tuning with LS-100 and LL-10 datasets, we double the number of masks to avoid overfitting.

**Optimization.** During the fine-tuning phase, we use the ADAM optimizer (Kingma & Ba, 2015) with a learning rate of $5 \cdot 10^{-5}$. The learning rate was selected between $5 \cdot 10^{-5}$ and $7 \cdot 10^{-5}$ after training the models for $5K$ steps. We use the exponential-decay function with the decay rate of $0.99$ as a learning rate scheduler. For LL-10, LS-100 and LS-960, we fine-tune the models for a maximum of $30K$, $100K$ and $400K$ steps respectively. We fine-tune all models with the batch size of $24$ accounting for 3 step gradient accumulation used with a batch size of $8$. We use the same parameters while fine-tuning with LS-960, LS-100 and LL-10 datasets.

### 3.5 DECODING AND LANGUAGE MODEL

For decoding, we use a CTC beam search decoder with and without language model (LM) to evaluate our models. A 4-gram LM trained on the LibriSpeech corpus is considered for decoding with the LM. We use beam width of 50 to measure the model performance with and without LM. We note that this beam width is smaller than higher performing systems, but we consider is a good trade-off with latency at deployment time. We tune the weights of the LM for the range of $[0.1, 2.0]$ with an interval of $0.1$ and a word insertion penalty for the range of $[-0.5, 0.5]$ with an interval of $0.05$.

### 3.6 METRICS

For ASR quality we consider the word error rate (WER) evaluation across various datasets, and for resource efficiency metrics we consider GPU hours and wall clock hours, as well as factors that influence these such as number of GPUs and batch size.

### 3.7 BASELINE

Past studies have proposed variations around the idea of learning speech representations by predicting the content of unseen regions (Ling et al., 2020; Ling & Liu, 2020; Liu et al., 2020b; Chi et al., 2021). On the other hand, recent studies have shown the potential of SSL pre-training by contrasting the target unseen frame with randomly sampled ones (Baevski et al., 2019). Similarly, studies in (Baevski et al., 2020; Hsu et al., 2021) have proposed to predict discrete targets of masked regions as the SSL training objective. As shown in (Hsu et al., 2021), Wav2Vec-2.0 and HuBERT approaches have demonstrated a great performance by outperforming all other reconstruction or contrastive alternative approaches. However, we only add Wav2Vec-2.0 as a baseline in our experiments since Wav2Vec-2.0 BASE model's performance (2.7 / 7.9, 3.4 / 8.0 on dev-clean/other, test-clean/other) is very similar to HuBERT BASE model's performance (2.7 / 7.8, 3.4 / 8.1 on dev-clean/other, test-clean/other).

### 3.8 SCOPE

It is worth noting that given our focus is on the relative merits between contrastive and non-contrastive methods, we restricted the HPO to make this exploration feasible. In particular, we did

not do extensive HPO on: (i) data augmentation layers, namely spec augment, (ii) batch size scaling which gives better results on contrastive rather than non-contrastive approaches, and (iii) static loss scaling parameters (i.e., $w_M$ and $w_U$). Nevertheless, trained on LibriSpeech dataset, our baselines have less than 1.5% absolute WER (on dev/test-clean splits) difference only when compared with SOTA solutions, e.g., Wav2Vec-2.0 (Baevski et al., 2020) which are the result of extensive HPO.

# 4 RESULTS

## 4.1 QUALITY COMPARISON OF NON-CONTRASTIVE AND MASK BASED CONTRASTIVE SSL

In order to examine the potential of our non-contrastive SSL method for learning effective speech embeddings, we first compare its performance, for the downstream ASR task, with the SOTA contrastive SSL method Wav2Vec-2.0 (Baevski et al., 2020). We perform this evaluation by fine-tuning the pre-trained models in two settings. Firstly, with a *high-resource setting* where large quantities of labeled speech are available. This enables us to compare the effectiveness of speech embeddings learned with contrastive and non-contrastive SSL methods. Secondly, with a *low-resource setting* where the amount of labeled data is limited. This helps us to compare the usefulness of speech embeddings, learned with the two methods, for improving low resource settings.

**Evaluation with High-Resource Labeled Data.**  We fine-tune the embedding models, pre-trained using W2V-2 and Non-Contrastive methods (discussed in Section 3.3), with LS-960 dataset. Additionally, we also fine-tune the pre-trained W2V-2 model with Non-Contrastive SSL approach to explore the performance of models when trained sequentially with both approaches (i.e., *Sequentially Combined* method). Though this approach is computationally inefficient, we investigate it to examine the potential of sequentially combined pre-training when resources are not a constraint or when a pre-trained model is available and we would like to further improve it.

Table 1 presents the ASR model performance in terms of WER (c.f. Appendix for character error rate performance). The results for these evaluations show *our proposed Non-Contrastive SSL method outperforms the SOTA W2V-2 approach for speech representation learning by achieving up to 6% relative WER improvement on dev/test-clean splits of LibriSpeech dataset.*

At the same time, our results show the *Sequentially Combined SSL approach significantly boosts the performance of downstream ASR task and achieves up to 12% relative WER improvement on dev/test-clean splits.* Conclusively, if the priority is to decrease WER rather than training time, Sequentially Combined SSL method is observed to yield the best results.

**Evaluation with Low-Resource Labeled Data.**  To evaluate the performance in low resource settings, we considered fine-tuning the W2V-2 and Non-Contrastive pre-trained models with LS-100 and LL-10 datasets (discussed in Section 3.2.

Table 2 presents the ASR model performance on low resource settings in terms of WER on LibriSpeech dev and test splits (additional character error rate performance reported in Appendix). Our results show that even on the low resource settings, the speech embeddings learned on unlabeled data with our non-contrastive SSL method are significantly more effective than the ones learned with contrastive SSL method. Specifically, *we observe up to 12% relative WER improvement on dev/test-clean sets on both LS-100 and LL-10 datasets*. Note that we did not perform HPO for the evaluation with LS-100 and LL-10, instead we used the optimal ones found for LS-960 evaluations. This is reflected in the quality performance compared to that reported in (Baevski et al., 2020). Nevertheless, we do see an improvement in terms of WER performance of *Non-Contrastive* SSL compared to contrastive *W2V-2* method.

## 4.2 COMPUTATIONAL AND RESOURCE EFFICIENCY

There has been great progress in the improvement of ASR systems which are founded through modeling advancements (e.g., Conformer (Gulati et al., 2020)), pre-training and learning methodologies (e.g.,Wav2Vec-2.0 (Baevski et al., 2020)), data augmentations (e.g., SpecAugment (Park et al., 2019)), and so on. Nevertheless, training time and the requirements for a huge number of resources remain a significant barrier to scale up these improvements for a wider number of practitioners, supported languages, latency requirements and deployment scenarios.

Table 1: Model performance in terms of WER on the Librispeech dev/test sets when fine-tuned on LS-960. Our *Non-Contrastive* SSL method outperforms the SOTA *W2V-2* approach (Baevski et al., 2020). *Sequentially Combined* SSL approach further boosts the performance of *W2V-2* model.

| Model | LM | dev | | test | |
|---|---|---|---|---|---|
| | | clean | other | clean | other |
| *W2V-2* | None | 4.33 | 11.21 | 4.47 | 11.36 |
| *Non-Contrastive* | None | 4.01 | 11.26 | 4.29 | 11.15 |
| *Sequentially Combined* | None | 3.81 | 11.11 | 4.01 | 11.00 |
| *W2V-2* | 4-gram | 2.99 | 7.97 | 3.30 | 8.10 |
| *Non-Contrastive* | 4-gram | 2.88 | 7.87 | 3.22 | 8.07 |
| *Sequentially Combined* | 4-gram | 2.83 | 7.90 | 3.20 | 8.10 |

Table 2: Model performance in terms of WER on the Librispeech dev/test sets when fine-tuned on LS-100 and LL-10. Our *Non-Contrastive* SSL method achieves marginal improvement in WER as compared to the SOTA *W2V-2* approach.

| Model | Fine-tuning Data | LM | dev | | test | |
|---|---|---|---|---|---|---|
| | | | clean | other | clean | other |
| *W2V-2* | 10 | None | 12.18 | 24.95 | 12.58 | 25.63 |
| *Non-Contrastive* | 10 | None | 12.17 | 24.17 | 12.19 | 24.81 |
| *W2V-2* | 10 | 4-gram | 6.65 | 17.22 | 7.04 | 17.76 |
| *Non-Contrastive* | 10 | 4-gram | 6.38 | 16.45 | 6.64 | 17.01 |
| *W2V-2* | 100 | None | 6.32 | 19.20 | 6.65 | 19.15 |
| *Non-Contrastive* | 100 | None | 6.42 | 18.47 | 6.84 | 18.30 |
| *W2V-2* | 100 | 4-gram | 4.03 | 13.64 | 4.51 | 13.71 |
| *Non-Contrastive* | 100 | 4-gram | 4.04 | 13.25 | 4.45 | 13.00 |

Due to this reason, in this section we investigate the computational and resource efficiency of non-contrastive versus contrastive SSL approaches for speech representation learning. Table 3 shows the quantification of the GPU hours required for pre-training with these SSL methods, and Table 4 presents the batch size requirements which is a key enabler of allowing for better embedding models at the cost of increased memory requirement.

Table 3 shows that *indeed in SSL for ASR, training time can be decreased via non-contrastive SSL (as proposed in this paper) rather than contrastive W2V-2 by as much as 2376 / 1064 = 2.23 times.* Furthermore, Table 3 also highlights that non-contrastive SSL method requires fewer GPUs when compared with contrastive W2V-2, which can be seen as a consequence of the required batch size in each approach as we discuss next.

Inline with the previous study (Zbontar et al., 2021), where contrastive approaches have been found to require much larger batch sizes than non-contrastive approaches. Our experimental setup, as described in Table 4, also highlights *fewer required GPUs and smaller batch size in non-contrastive (960 secs) as opposed to contrastive (5376 secs) approaches, allowing for training on memory constrained GPUs.*

Finally, it is worth noting that if the priority is to decrease WER rather than training time or GPU resources, lowest WER can be achieved by sequentially combined contrastive followed by non-contrastive training. Although, we have already discussed WER improvements from this approach in the previous subsection, we would like to highlight that the increase in computation time and batch size cost is only of a small fraction as can be seen in Tables 3 and 4.

### 4.3 Impact of Batch Size and Sequence Length on Computation Time

In this subsection, we perform an ablation study to understand the impact of the batch size and sequence length on training time. In particular, having as a reference the setting considered in

Table 3: Computation time for pre-training with different SSL approaches. Compared to *W2V-2*, our *Non-Contrastive* approach requires $2.23\times$ less GPU hours. With a fraction of extra computation time, we can further boost the performance of *W2V-2* using *Sequentially Combined* method.

| Model | GPU Hours | No. of GPUs | Wall Clock Hours |
|---|---|---|---|
| *W2V-2* | 2457 | 64 | 38 |
| *W2V-2* | 2376 | 8 | 297 |
| *Non-Contrastive* | 1064 | 8 | 133 |
| *Sequentially Combined* | 3040 (2376+664) | 8 | 380 (297+83) |

Table 4: Batch size requirements for pre-training with different SSL approaches. Our results show that *Non-Contrastive* SSL method requires smaller batch sizes as compared to *W2V-2* approach. For non-contrastive method we use the batch size per GPU of 120 secs by leveraging the gradient accumulation of 3 steps each with batch size of 40 secs.

| Model | No. of GPUs (V100's) | Batch size/GPU (secs) | Batch size (secs) |
|---|---|---|---|
| *W2V-2* | 64 | 84 | 5376 |
| *W2V-2* | 8 | 84 (x8) | 5376 |
| *Non-Contrastive* | 8 | 40 ($\times3$) | 960 |
| *Sequentially Combined* | 8 | 40 ($\times3$) | 960 |

Table 5: Training cost (in terms of GPU hours) for Non-Contrastive SSL method with different batch sizes and sequence lengths. Results are computed when training on 8 V100 GPUs with gradient accumulation of 3 steps. We observe that increasing the batch size results in a higher computation time than increasing the sequence lengths. Similarly, the impact on WER is reported in the Appendix.

| Batch Size | GPU Hours | | |
|---|---|---|---|
| | Seq Len 3s | Seq Len 5s | Seq Len 7s |
| **24** | 704 | 1064 | 1483 |
| **48** | 1216 | OOM | OOM |
| **72** | 1696 | OOM | OOM |

Wav2Vec-2.0, namely batch size 6 with approx. 15s in each sample resulting in a lower amount of audio per batch of 1.4min. We expand our measurements from a single setting of batch size 24 with 5s in each sample with yields 2min of audio per batch to different configurations of batch sizes 24, 48 and 72 as well as sequence lengths of 3s, 5s and 7s. We limit the exploration to maximum batch size of 72 and sequence length of 7s as allowed by our infrastructure (with NVIDIA V100 GPUs) without increasing the gradient accumulation steps (fixed at three as in other experiments).

As shown in Table 5, we observe that increasing the batch size results in a higher computation time than increasing the sequence lengths. This may be due to GPU contention at higher batch sizes, or that depending on the configuration of batch size and sequence length different kernels for the different operations are being selected by the backend. Moreover, given that the non-contrastive approach is based on a siamese network format, with exponential moving average, there are twice the weights in memory as well as twice the number of forward passes which limits how much we can increase the batch size and the sequence lengths. Therefore, even with three step gradient accumulation (as used in all other experiments in the paper) we can see in the table that the more demanding settings result in out of memory (OOM) on V100 GPUs having 32 GB memory.

## 5 CONCLUSION

This work made two contributions towards resource efficient speech representation learning for ASR. First, we investigated the potential of replacing established contrastive SSL approaches (e.g., Wav2Vec-2.0 (Baevski et al., 2020)) with a non-contrastive SSL method to reduce training time and GPU resources needed for training ASR systems. Second, we extended existing image based non-contrastive Barlow–Twins (Zbontar et al., 2021) method for speech representation modeling and more generally to arbitrary time varying data. Numerical results demonstrated a training speed up of 2.23 times as well as improved model performance (i.e., lower WER) via the proposed non-contrastive as opposed to contrastive approaches. Finally, our results demonstrated the benefits of sequentially combining contrastive and non-contrastive methods to further boost ASR performance.

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

## A  Appendix

MODEL PERFORMANCE IN TERMS OF CER

To examine the ASR model performance in terms of label prediction on low and high resource settings, we compute character error rate (CER) on LibriSpeech dev and test splits. The results for these evaluations are presented in Tables 6 and 7.

Table 6: Model performance in terms of CER on the Librispeech dev/test sets when fine-tuned on LS-960. Both *Non-Contrastive* and *Sequentially Combined* SSL methods outperform the SOTA *W2V-2* approach with a marginal difference.

| Model | LM | dev | | test | |
|---|---|---|---|---|---|
| | | clean | other | clean | other |
| *W2V-2* | None | 1.30 | 4.38 | 1.31 | 4.30 |
| *Non-Contrastive* | None | 1.23 | 4.40 | 1.26 | 4.24 |
| *Sequentially Combined* | None | 1.16 | 4.41 | 1.19 | 4.20 |
| *W2V-2* | 4-gram | 1.02 | 3.64 | 1.08 | 3.53 |
| *Non-Contrastive* | 4-gram | 0.99 | 3.67 | 1.05 | 3.59 |
| *Sequentially Combined* | 4-gram | 0.97 | 3.77 | 1.05 | 3.65 |

Table 7: Model performance in terms of CER for a more fine-grained analysis in terms of label prediction on the Librispeech dev/test sets when fine-tuned on LS-100 and LL-10. Our *Non-Contrastive* SSL method achieves marginal improvement in CER as compared to the SOTA *W2V-2* approach.

| Model | Fine-tuning Data | LM | dev | | test | |
|---|---|---|---|---|---|---|
| | | | clean | other | clean | other |
| *W2V-2* | 10 | None | 3.90 | 10.15 | 3.99 | 10.12 |
| *Non-Contrastive* | 10 | None | 3.83 | 9.80 | 3.85 | 9.72 |
| *W2V-2* | 10 | 4-gram | 2.70 | 8.61 | 2.82 | 8.59 |
| *Non-Contrastive* | 10 | 4-gram | 2.62 | 8.5 | 2.63 | 8.40 |
| *W2V-2* | 100 | None | 1.96 | 7.98 | 2.09 | 7.66 |
| *Non-Contrastive* | 100 | None | 1.99 | 7.57 | 2.12 | 7.29 |
| *W2V-2* | 100 | 4-gram | 1.47 | 6.80 | 1.68 | 6.60 |
| *Non-Contrastive* | 100 | 4-gram | 1.51 | 6.61 | 1.64 | 6.28 |

IMPACT OF AUDIO SEQUENCE LENGTH DURING PRE-TRAINING ON CER AND WER

We focused our numerical experiments on investigating the advantages of the proposed method in terms of WER, batch size and GPU hours required with batches formed of 5 seconds audio samples. In addition to the GPU hours ablation study with 24, 48 and 72 batch sizes and 3 and 7 seconds audio clips, below we report the effect of sequence lengths on CER and WER for 3 and 7 seconds. In particular, Table 8 shows that shorter sequences such as 3 seconds do have an impact on the quality of the learned representations, whereas longer sequences of 5 and 7 seconds led to similar and much better learned representation with small variations in the results.

Table 8: Effect of sequence lengths on model performance. The results show that shorter sequences such as 3 seconds negatively affect the quality of the learned representations. Whereas, longer sequences of 5 and 7 seconds led to similar and much better learned representation with small variations in the results.

| Sequence Length | Metric | dev | | test | |
|---|---|---|---|---|---|
| | | clean | other | clean | other |
| 3 | CER | 1.68 | 4.74 | 1.68 | 4.61 |
| 5 | CER | 1.23 | 4.40 | 1.26 | 4.24 |
| 7 | CER | 1.28 | 4.34 | 1.31 | 4.21 |
| 3 | WER | 5.63 | 12.47 | 5.77 | 12.63 |
| 5 | WER | 4.01 | 11.26 | 4.29 | 11.15 |
| 7 | WER | 4.12 | 11.18 | 4.33 | 11.12 |

SEQUENTIALLY COMBINED SSL MODEL PERFORMANCE ON LOW-RESOURCE SETTINGS

To complement the results for low-resource settings presented in Table 2, we present the results of ASR model performance when pre-trained with Sequentially Combined method compared to other approaches in Table 9. The models were fine-tuned on LL-10 and LS-100 datasets. The evaluation is performed on the dev and test splits. The results show that sequentially combined SSL method achieves marginal WER improvement as compared to the SOTA W2V-2 and Non-contrastive SSL approaches.

Table 9: Sequentially combined pre-trained model's performance in terms of WER on the Librispeech dev/test sets when fine-tuned on LS-100 and LL-10. Sequentially combined SSL method achieves marginal improvement in WER as compared to the SOTA W2V-2 and Non-contrastive SSL approaches.

| Model | Fine-tuning Data | LM | dev | | test | |
|---|---|---|---|---|---|---|
| | | | clean | other | clean | other |
| W2V-2 | 10 | None | 12.18 | 24.95 | 12.58 | 25.63 |
| Non-Contrastive | 10 | None | 12.17 | 24.17 | 12.19 | 24.81 |
| Sequentially Combined | 10 | None | 11.97 | 24.18 | 12.27 | 24.91 |
| W2V-2 | 100 | None | 6.32 | 19.20 | 6.65 | 19.15 |
| Non-Contrastive | 100 | None | 6.42 | 18.47 | 6.84 | 18.30 |
| Sequentially Combined | 100 | None | 6.48 | 18.03 | 6.81 | 18.01 |

