# OpenReview forum: "Resource Efficient Self-Supervised Learning for Speech Recognition"
_ICLR.cc/2023/Conference — Submitted to ICLR 2023_

### Official Review · Reviewer_t7qA · 2022-10-24

**Confidence:** 4
**Correctness:** 3
**Technical Novelty And Significance:** 3
**Empirical Novelty And Significance:** 3
**Recommendation:** 6

**Clarity, Quality, Novelty And Reproducibility:**

— Page 7 – “discussed in Section ??”

— In the abstract, the authors say “a combination of both masked based SSL and non-contrastive SLL” improves the ASR performance. Here, ‘masking-based SSL’ actually means contrastive learning solution (e.g., wav2vec 2.0)? I think masking is also used for non-contrastive solutions.

— In table 1&2, the ‘Iteratively combined’ solution seems to be a misleading name. Do authors really try to iteratively apply contrastive and non-contrastive training? I think it probably means fine-tuning a contrastive model with non-contrastive loss.

— In Table 5, the resource consumption of “iteratively combined” should be the sum of both wav-2 and non-contrastive.

— In page 5, the authors claim that they do not perform any pre-processing; It seems in wav2vec, they do use mean/std normalization.


**Strength And Weaknesses:**

The strength:
— To my knowledge, this is the first work that applies Barlow-Twins methodology to learn representation for ASR; The adoption of BT into sequential representation learning is not trivial. Previous, I only find BT is applied for speaker recognition.

— The authors show that the proposed methods are more resource efficient and achieve comparative performance. A) It improves convergence, b) it reduces training time, c) it significantly reduces GPU training times, d) it requires smaller batch size thus reducing memory requirements.

— The authors also found that combining the proposed method with a contrastive learning approach is helpful.


The draft can be improved with some clarification and extra ablation studies:
— Same as wav2vec 2.0, the proposed methods apply masks to the CNN-extracted features; Here, the mask is only used to distort the input (to my understanding), thus the authors can explore ways of distortions other than applying masks.

— Regarding Time Unrolling and time merging losses: The proposed method chunks the audio into 5-seconds. It seems that for longer input or when using larger batch size/projection dimension, the calculation of the two losses can be much time-consuming (i.e., slower). Also, what’s the relative importance of the two losses? Maybe an ablation study with different w_U and w_M (static loss) could be interesting.

— The authors show that further fine-tune wav2vec 2.0 with the proposed methods achieves better performance than both wav2vec 2.0 and the proposed non-contrastive training. What if the authors first train with the non-contrastive loss and then fine-tune with contrastive loss?

— The “iteratively combined” solution is only applied to LS-960; It would be good to see the "iteratively combined" solution also applied on low-resource scenarios, and see its relative improvement.


**Summary Of The Paper:**

While wav2vec-style contrastive learning has shown to be very successful for ASR, it requires a lot of resources and time for training. In the vision domain, Barlow Twins, a solution that naturally avoids collapse, has shown to be able to achieve better (or competitive) performance compared with contrastive learning (e.g., SimCLR) while using much smaller batch size. However, Barlow Twins style training is under-explored in the ASR/audio domain.

The authors applied revised Barlow Twins training to the wav2vec-2 framework (CNN feature extractor + Transformer), and achieved competitive performance compared to wav2vec 2.0, while reducing training time, GPU usage and improving convergence.


**Summary Of The Review:**

At this moment, I’m inclined to recommend this paper, though the paper could be in a better shape with more clarification and ablation study.

---

> ### Author Response · Authors · 2022-11-16
> **Replies part-2**
>
>
>
> **The authors show that further fine-tune wav2vec 2.0 with the proposed methods achieves better performance than both wav2vec 2.0 and the proposed non-contrastive training. What if the authors first train with the non-contrastive loss and then fine-tune with contrastive loss?**
>
> We agree that there could be multiple ways to combine the contrastive and non-contrastive training such as C->NC (which we report on), NC->C, versus an iterative sequence like C->NC->C… or NC->C->NC…, including the exploration of optimal schedulers to switch between contrastive and non-contrastive approaches. But we leave these for further exploration as these are not in the scope of this work.
>
> In particular, our priority for this work has been threefold: Firstly, we aimed to show that non-contrastive approaches are worth exploring in the audio domain which is currently dominated by masking-based and contrastive methods. Secondly, not only are non-contrastive approaches competitive from a WER point of view, they produce similar WER at lower wall clock time for the same computational resources. Thirdly, having established the fact that there are more than one type of methods for SSL ASR, we began an exploration on potential advantages of iteratively combining them for decreased WER as compared to using only one of them.
>
> **The “iteratively combined” solution is only applied to LS-960; It would be good to see the "iteratively combined" solution also applied on low-resource scenarios, and see its relative improvement.**
>
> We agree with the reviewer that it’d be optimal to include iteratively combined results for low-resource scenarios as well. However, looking at the set of all reviewer’s suggestions (including this reviewer request for additional information of how the proposed time unrolling and time merging losses would be affected under differently sized inputs), we’ve given priority to providing gpu hours for other more critical experiments. If time perhaps however, we’ll run these experiments as well.
>
>
>
> **Page 7 – “discussed in Section ??”**
>
> Thank you for pointing this out. We’ve corrected the reference.
>
> **In the abstract, the authors say “a combination of both masked based SSL and non-contrastive SLL” improves the ASR performance. Here, ‘masking-based SSL’ actually means contrastive learning solution (e.g., wav2vec 2.0)? I think masking is also used for non-contrastive solutions.**
>
> We agree that this sentence adds some confusion. We will rephrase it as follows:
> `We further demonstrate that a combination of both masking-based contrastive and non-contrastive SSL improves the ASR performance....`
>
>
> **In table 1&2, the ‘Iteratively combined’ solution seems to be a misleading name. Do authors really try to iteratively apply contrastive and non-contrastive training? I think it probably means fine-tuning a contrastive model with non-contrastive loss.**
>
> The motivation for the phrasing “Iteratively combined” came from the idea of improving ASR WER performance by iteratively combining contrastive and non-contrastive approaches (such as C->NC->C… or NC->C->NC…) of which in this paper we have realized the C->NC case. Given this, we agree that in the current experiments this phrasing may be improved to “Sequentially Combined”, which we’ve changed throughout the text.
>
>
> **In Table 5, the resource consumption of “iteratively combined” should be the sum of both wav-2 and non-contrastive.**
>
> Thank you very much for this comment. We see that during the last minute changes before submission we removed a sentence detailing this information from the caption. For maximum clarity, we will modify the table to report the sum of Wav2Vec-2.0 plus non-contrastive like so: 3121 (2457 + 664). In addition, we will also clarify this in the caption.

---

> > ### Author Response · Authors · 2022-11-18
> > **Update on results for low-resource settings**
> >
> > We're happy to report that we managed to run sequentially combined (previously iteratively combined) for the low-resource setting which we've added to the Appendix, e.g., Table 9. In particular, we see again the benefit of sequentially combined training as opposed to W2V2.
> >
> >
> >
> > Model | Fine-tuning Data | dev-clean | dev-other | test-clean | test-other
> > ---------------|:----------------:|:---------------:|:---------------:|:---------------:|:---------------:
> > **W2V-2** | LL-10 | 12.18 | 24.95 | 12.58 | 25.63
> > **Non-Contrastive** | LL-10 | 12.17 | 24.17 | 12.19 | 24.81
> > **Sequentially Combined** | LL-10 | 11.97 | 24.18 | 12.27 | 24.91
> > **W2V-2** | LS-100 | 6.32 | 19.20 | 6.65 | 19.15
> > **Non-Contrastive** | LS-100 | 6.42 | 18.47 | 6.84 | 18.30
> > **Sequentially Combined** | LS-100 | 6.48 | 18.03 | 6.81 | 18.01

---

> ### Author Response · Authors · 2022-11-16
> **Replies part-1**
>
> We thank the reviewer for helpful comments and suggestions. In particular, they led to the following changes:
> - Added discussion on design choice of sequence length of 5secs.
> - Improve readability and address some typos.
>
>
> **The draft can be improved with some clarification and extra ablation studies: — Same as wav2vec 2.0, the proposed methods apply masks to the CNN-extracted features; Here, the mask is only used to distort the input (to my understanding), thus the authors can explore ways of distortions other than applying masks.**
>
> We agree with the reviewer that our approach uses masking similar to Wav2Vec-2.0. However, in Wav2Vec-2.0 these masks are used to perform contrastive learning (similar to masked language modeling), whereas, in our approach these masks are used to simply augment the input to the encoders.
>
> We chose SpecAugmentation for two reasons. Firstly it’s very popular and in general an effective strategy in the domain of ASR. Secondly, we chose the particular version of SpecAugmentation with vertical bars (i.e., for each masked timestamp, all features are masked), but not with horizontal bars (i.e., to mask specific features across all time steps). This setup more closely matches the one in Wav2Vec-2.0, which limits the confounding aspects when comparing between the two approaches.
>
> We understand that there could be other approaches for augmenting the input. One of the latest directions is to perform modality agnostic and policy based augmentation in latent space [1, 2] that could be combined with our approach. We see value in exploring different forms of data augmentation, therefore a future direction to extend this study could be to explore other augmentation schemes.
>
> [1] Wang et al. G-Augment: Searching For The Meta-Structure Of Data Augmentation Policies For ASR. arXiv preprint arXiv:2210.10879.
>
> [2] Cheung et al. Modals: Modality-agnostic automated data augmentation in the latent space. In CLR’21.
>
>
> **Regarding Time Unrolling and time merging losses: The proposed method chunks the audio into 5-seconds. It seems that for longer input or when using larger batch size/projection dimension, the calculation of the two losses can be much time-consuming (i.e., slower). Also, what’s the relative importance of the two losses? Maybe an ablation study with different w_U and w_M (static loss) could be interesting.**
>
> Regarding computational cost for the losses, we agree that it increases with larger batch sizes, sequence lengths and projection sizes. In particular, decreasing computational cost was a motivating factor to exclude the third alternative of creating a loss term based on a [T, T] matrix, and focus instead on time unrolling and merging losses. For the latter two, the computational requirements are lower given that time unrolling loss requires the creation of a [F, F] matrix where each component is the result of an inner product between vectors of size B * T, and time merging loss needs the creation of a [B, B] matrix wherein each element is the result of a dot product between vectors of size T * F.
> To investigate the impact of possible batch sizes and sequence length on time unrolling and merging training time, we conducted the following ablation study. Here, in particular we could increase the sequence length from 5 seconds to 7 seconds while keeping the same batch size of 24 with a 1.4x increase in GPU hours, as well as consider other alternatives as shown in the table.
>
> &nbsp; | Seq Length: 3s | Seq Length: 5s | Seq Length: 7s
> ---------------|:----------------:|:---------------:|:---------------:
> **Batch Size 24** | 704 hrs | 1064 hrs | 1483 hrs
> **Batch Size 48** | 1216 hrs | OOM | OOM | OOM
> **Batch Size  72** |1696 hrs | OOM | OOM | OOM
>
>
> Regarding the relative merits between the time unrolling and merging losses, we have observed that during the pre-training phase, if we remove either of these losses then the model collapses within a few steps. Moreover, we have observed that the training with static scaling of losses could not converge as compared to dynamic loss scaling approach. This made it even more difficult to perform such an ablation study. Perhaps the static loss scaling required more HPO to be done as we tried with possible values of w_M and w_U in [0.1, 0.5, 1.0]. We’ll include this information in the paper as well.

---

### Official Review · Reviewer_5Dwq · 2022-10-25

**Confidence:** 4
**Correctness:** 3
**Technical Novelty And Significance:** 3
**Empirical Novelty And Significance:** 3
**Recommendation:** 5

**Clarity, Quality, Novelty And Reproducibility:**

The paper is well-written and easy to follow. The paper extends the non-contrastive self-supervised training to the audio modality which is novel.

**Strength And Weaknesses:**

Strengths:

(i) Paper clearly shows the extension of Barlow-Twins methods for audio modality with intuitive losses.
(ii) The empirical results show the method's efficiency (compute resources) compared to the Wav2vec2-like pre-training.

Weakness:

(i) In the section (augmentation in latent space), it says that masks are applied with a probability of 0.05 and mask length of 10 for the target model and 0.1 and 20 for the online model.
- What is the fraction of input unchanged in the output of both the target model and online model representations?
- Time unrolling loss compares similarity across time-frames for the same frames and if most of the temporal frames are unchanged then the solution would be trivial. I am curious in understanding the effect of augmentation on the overall objective, which is not described in the paper.
- In images, the augmentations(cropping, resizing, flipping, etc.) are more meaningful, which is not a luxury in speech. Due to the fixed audio duration for each batch (5 ms), perturbations in the frequency domain cannot be carried out, which further limits the augmentation space.

(ii) I do not see the comparison of static and dynamic scaling for the loss functions. Is the idea of using a stop gradient for dynamic loss novel? If not the citation for that is missing.

(iii) Paper states that for the pre-training the audio utterances are cropped into 5 seconds. (Wav2vec2 uses 15.6 seconds).
- Was this design choice considering the GPU size?
- What is the effect of having longer utterances on the overall pre-training objective?
- Because Non-contrastive learning methods don't require larger batch sizes, does having longer utterances help or deteriorate the pre-training performance?

(iv) I find *Iteratively Combined* baseline overtly powerful because of the number of customization steps (400K for wav2vec2 + 250k for the Non-contrastive method), as compared to W2V2 and Non-Contrastive method.
- I would suggest running each of the methods to a fixed number of steps and then comparing. Since the number of parameters are the same, this would give a more accurate picture of the efficacy of each of these methods.

(v) Results of other self-supervised methods such Hubert, Decoar, etc. should be placed to understand the relative strength of the non-contrastive method.

(vi) Ablation of both the Time Unrolling and Time Merging loss is missing. Very curious to know what is the impact of time merging loss on the overall objective.

Minor:
(i) A reference missing in line 3 of the `Evaluation with High-Resource Labeled Data` paragraph.





**Summary Of The Paper:**

This paper describes non-contrastive learning based self-supervised pre-training approach for the speech recognition task. As opposed to the popular methods such as Wav2vec2, Hubert, etc. which use contrastive learning for pre-training, this paper describes how non-contrastive methods such Barlow-Twins for the audio pre-training.

The main contribution of the paper is:
(i) Extend Barlow-Twins (Siamese Network) training for audio modality (with additional time axis) with new loss functions for that additional axis.

**Summary Of The Review:**

Although the paper has some interesting ideas for improving the efficiency of self-supervised training for speech recognition, it lacks an extensive understanding of the various design choices made in the paper. While the paper is interesting, by adding certain key details it would be easy to understand the positioning of the non-contrastive methods in the self-supervised pre-training landscape.

---

> ### Author Response · Authors · 2022-11-18
> **Replies part-2**
>
>
> **I find Iteratively Combined baseline overtly powerful because of the number of customization steps (400K for wav2vec2 + 250k for the Non-contrastive method), as compared to W2V2 and Non-Contrastive method. I would suggest running each of the methods to a fixed number of steps and then comparing. Since the number of parameters are the same, this would give a more accurate picture of the efficacy of each of these methods.**
>
> Reflecting on the reviewer suggestion of running each approach for a fixed number of steps, we would say the following. Contrastive and non-contrastive training objectives are significantly different, and therefore we would argue there is not a strong case to expect that there would be a benefit of training for the same number of steps in an iterative approach. Rather we believe a better strategy would be to impose a maximum number of steps, leverage early stopping to roll-back to the best checkpoint before switching to the next iterative approach. We already see this in the table below where in particular we see that the second step of an interactive sequence C->NC would benefit from starting from 359K rather than 400K.
>
> Another alternative would be to use a scheduler that trains the network with a sequence of contrastive and non-contrastive (such as C->NC->C… or NC->C->NC…) which might further improve the performance of the model with potentially less number of steps. However, we leave this exploration for future work as the scope of this paper is to show that non-contrastive approaches are worth exploring in the audio domain to achieve similar or even better performance with lower wall clock time for the same computational resources as compared to existing masking-based and contrastive methods.
>
> Fine-tuning Data | Steps (with early stopping) | Max Steps
> ---------------|:----------------:|:---------------:
> LL-10 | 26K | 30K
> LS-100 | 83K | 100K
> LS-960 | 359K | 400K
>
>
>
> **Results of other self-supervised methods such Hubert, Decoar, etc. should be placed to understand the relative strength of the non-contrastive method.**
>
> Our motivation to focus on Wav2Vec-2.0 rather than also additionally HuBERT was grounded on the fact that previous works have shown similar performance of these models across different datasets. In particular, Wav2Vec-2.0 BASE model’s performance is 2.7 / 7.9, 3.4 / 8.0 on dev-clean/other, test-clean/other, which is very similar to HuBERT BASE model’s performance (i.e., 2.7 / 7.8, 3.4 / 8.1 on dev-clean/other, test-clean/other). Therefore, we did not produce another similar baseline (such as HuBERT) on our training pipeline.
>
> Moreover, other methods such as DeCoAR, SlimIPL and DiscreteBERT have shown to be performing worse than Wav2Vec-2.0 BASE and HuBERT BASE models, so we did not reproduce these baselines on our training pipeline. We will add this discussion  in the  Evaluation Settings section for clarity.
>
> **Ablation of both the Time Unrolling and Time Merging loss is missing. Very curious to know what is the impact of time merging loss on the overall objective.**
>
> We have observed that during the pre-training phase, if we remove either of these losses then the model collapses within a few steps. Moreover, we have observed that the training with static scaling of losses could not converge as compared to dynamic loss scaling approach. This made it even more difficult to perform such an ablation study.  However, we agree with the reviewer that the paper would benefit from saying this explicitly, so we will add this to the results section.
>
> **A reference missing in line 3 of the Evaluation with High-Resource Labeled Data paragraph.**
>
> Thanks for pointing this out. We will fix this.

---

> > ### Comment · Reviewer_5Dwq · 2022-11-30
> > **Response to Author Comments**
> >
> > Dear Authors,
> >
> > I went through all the comments my fellow reviewers pointed out and the author's response to all the reviews. I thank the authors for conducting additional experiments demonstrating the technique's usefulness in low-resource settings and ablation on the sequence length.
> >
> > As pointed out by reviewers (5kcU and hCQK), data2vec is relevant and an important comparison that is missing in this work.
> >
> > After reading the data2vec paper and reviewer hCQK's comment, I also see that there is a discrepancy in the paper's version of the wav2vec2 versus the wav2vec2.0 reported in the original and data2vec paper. The current implementation of wav2vec2 is inferior to what is reported in the literature by a large margin as reported in Table 3.
> >
> > *Test-Other*
> >
> > Current implementation: 24.95 (10H fine-tuning with 4-gram LM)
> >
> > Wav2vec2 original paper: 9.5 ( 10H fine-tuning with 4-gram LM), both pre-trained on LS-960H.
> >
> > *Test-Clean*
> >
> > Current Implementation: 11.68 (10H fine-tuning with 4-gram LM)
> >
> > Wav2vec2 original paper: 4.3 ( 10H fine-tuning with 4-gram LM), both pre-trained on LS-960H.
> >
> > There is some more work required in terms of reproducing the correct baseline.

---

> > > ### Author Response · Authors · 2022-12-07
> > > **Reply**
> > >
> > > Regarding the first point, we have answered this above and copying our reply here for convenience.
> > >
> > > >Our viewpoint on this is that our work is about resource efficient training rather than decreasing WER to establish a new SOTA on LibriSpeech. In that line, we would like to clarify our viewpoint in detail as follows. From the training efficiency point of view, we note that like our paper, data2vec also employs a siamese network and learns in a non-contrastive fashion, but does so via feature masking and prediction between teacher and student rather than our proposed approach which instead brings together a Barlow–Twins like loss to the audio domain, to dramatically improve training efficiency. This is an important point for simplifying SSL training for resource efficiency as also highlighted by reviewer hCQK. Specifically, we’d like to highlight that even though data2vec decreases the required batch size from 1.6 hours (as in wav2vec2) to 1.05 hours, our approach decreases it by a much more significant margin to 0.26 hours, which in our mind speaks to the advantages of our method in terms of decreased required batch size. Also from a training time point of view, data2vec is much slower than our approach. In particular, data2vec performs the same number of steps (400K) as we do, but their batch size is much larger than ours, specifically 4x larger (1.05 / 0.26 = 4). Even though data2vec doesn’t report training time numbers. Given the fact that both data2vec and our approach employ (even if in a different manner) siamese networks but data2vec uses 4x larger batch size, indicates that data2vec is significantly slower to train than our approach.
> > >
> > > Regarding the second point about results on test-clean/other, it seems that the review is looking at the previous version of the paper. We have updated these results after performing some more HPO on LM decoding params.

---

> ### Author Response · Authors · 2022-11-18
> **Replies part-1**
>
> **In the section (augmentation in latent space), it says that masks are applied with a probability of 0.05 and mask length of 10 for the target model and 0.1 and 20 for the online model.**
> - **What is the fraction of input unchanged in the output of both the target model and online model representations?**
> - **Time unrolling loss compares similarity across time-frames for the same frames and if most of the temporal frames are unchanged then the solution would be trivial. I am curious in understanding the effect of augmentation on the overall objective, which is not described in the paper.**
>
> The first question was a little difficult to understand. Our interpretation of the reviewer’s question is: in practice what are the number and size of the masked frames on each spec augment layer in the target and online models? Assuming this is correct, our response is as follows: For the target model there are 2 mask spans of length 10, so total 20 masked frames (i.e., 400 ms). Whereas, for the online model there are 3 mask spans of length 30, so total 60 masked frames (i.e., 1200 ms).
>
> Regarding the effect of augmentation on the overall objective, trivial solutions are not encouraged by the time unrolling loss due to the following reason.
> Due to the self-attention layers in the Wav2Vec-2.0 model, a small localized change in one hidden representation of input to the encoder will change the output in a non-localized manner (i.e., that localized change in the input will be propagated globally while computing the output of the encoder). This means that even if the target and online models were the same (even though they are initialized with different weights in our method), the non-overlapped non-masked hidden representations time steps would result in different outputs in those time steps. Therefore, time unrolling loss would not compare similar time-frames in the output, thereby not encouraging a trivial solution.
>
> **I do not see the comparison of static and dynamic scaling for the loss functions. Is the idea of using a stop gradient for dynamic loss novel? If not the citation for that is missing.**
>
> Yes, the idea of using stop gradients for dynamic scaling of losses is novel. We have observed that the training with static scaling of losses could not converge as compared to dynamic loss scaling approach. Therefore, we could not add results for fine-tuning the models pre-trained with static loss scaling. Perhaps the static loss scaling required more HPO to be done as we tried with possible values of w_M and w_U in [0.1, 0.5, 1.0]. We have discussed this in section 3.3 (last paragraph): `We observed that the convergence with dynamic scaling of the loss (discussed in 2.2) was better than the static option, which motivated us to use it throughout our experiments`. We will clarify it further in the text.
>
> **Paper states that for the pre-training the audio utterances are cropped into 5 seconds. (Wav2vec2 uses 15.6 seconds).**
> - **Was this design choice considering the GPU size?**
> - **What is the effect of having longer utterances on the overall pre-training objective?**
>
> Given that the proposed non-contrastive approach, leverages a siamese network which requires twice the amount of memory for weights and twice the forward passes we had to limit the sequence length. From the point of view of selecting the (batch size, sequence length) tuple we conducted time measurement experiments to find a tuple that would take advantage of the V100 GPUs we used, and found that (24, 5 * 16000) was a good trade-off. In this regard, below is a table presenting training time for alternative options.
>
>
> &nbsp; | Seq Length: 3s | Seq Length: 5s | Seq Length: 7s
> ---------------|:----------------:|:---------------:|:---------------:
> **Batch Size 24** | 704 hrs | 1064 hrs | 1483 hrs
> **Batch Size 48** | 1216 hrs | OOM | OOM | OOM
> **Batch Size  72** |1696 hrs | OOM | OOM | OOM
>
> In addition to the reported GPU hours in the table above, we’ve also trained models under these different settings with inputs having sequence lengths of 3 and 7 seconds without further HPO (e.g., same learning rate and augmentation settings), and report the WER below. We again thank the reviewer for this comment which shows that shorter sequences such as 3 seconds do have an impact on the quality of the learned representations, whereas longer sequences of 5 and 7 seconds led to similar and much better learned representation with small variations in the results. We have added these results in the appendix of the paper.
>
> Seq Length | Metric | dev-clean | dev-other | test-clean | test-other
> ---------------|:----------------:|:---------------:|:---------------:|:---------------:|:---------------:
> **3** | CER | 1.68 |  4.74 |  1.68 | 4.61
> **5** | CER | 1.23 |  4.40 |  1.26 |  4.24
> **7** | CER | 1.28 |  4.34 |  1.31 |  4.21
> **3** | WER | 5.63 |  12.47 |  5.77 |  12.63
> **5** | WER | 4.01  |  11.26 |  4.29 |  11.15
> **7** | WER | 4.12 |  11.18 |  4.33 |  11.12

---

> ### Author Response · Authors · 2022-11-18
> **Summary of revision and replies**
>
> We thank the reviewer for helpful comments and suggestions. In particular, they led to the following changes:
> - Improved discussion of the impact of spec augmentation in the proposed time unrolling and merging losses.
> - Highlighted novelty of the dynamic loss component of the approach.
> - Added discussion on design choice of sequence length of 5secs.
> - Added discussion on the number of steps in each of the approaches when combined iteratively.
> - Added discussion on the combined value of the proposed time unrolling and merging losses as opposed to only one of them.

---

### Official Review · Reviewer_hCQK · 2022-10-25

**Confidence:** 5
**Correctness:** 2
**Technical Novelty And Significance:** 3
**Empirical Novelty And Significance:** 2
**Recommendation:** 5

**Clarity, Quality, Novelty And Reproducibility:**

**Clarity**
The paper is written well, however it could benefit with revision to simplify reading and remove repeated things in several places in the paper.

**Quality**
The idea itself is clear and well presented, however experimental setup and evaluation raise a bunch of questions which you can find below. Right now I have doubts about proper experimentation and efficiency metrics.

**Novelty**
The paper adapts known from vision Barlow-Twins loss while reducing the time dim in reshaping procedure. This reshaping is new component, the rest are known parts. However I am not sure why we should do time reduction and not some another dim.

**Reproducibility**
There are specified parameters in the paper on the training. Paper can benefit from revision where some details are specified too, e.g. how wav2vec 2.0 fine-tuning is exactly done to reproduce the prior paper.

**Detailed questions and suggestion to improve the text**
- Introduction: reference to wav2vec 2.0 in the context of absence of high quality large labeled data for many languages - it is strange usage of reference, maybe authors meant wav2vec multilingual paper then? Large curated unlabeled data - we have any more appeared since which actually gave huge speed up in development for SSL and semi-sup learning, e.g. LibriLight, VoxPopuli, Common Voice. For SSL in speech there are missing references to data2vec (Baevski, A., et al. Data2vec: A general framework for self-supervised learning in speech, vision and language, 2022, arXiv preprint arXiv:2202.03555), wav2vec-BERT (Chung, Y.A., et al. w2v-bert: Combining contrastive learning and masked language modeling for self-supervised speech pre-training, ASRU, 2021), autoregressive/non-autoregressive predictive coding e.g. from James Glass team, etc.
- Figure 1. It will be very clear if we show properly the matrices in time merging and unrolling same way as used in the text (e.g. transpose the matrix for merging) and also add dimensions to all matrices on the plot to see reshape operation right away there. This can simplify reading the image and allow clear following it.
- do authors do only SpecAugment on time domain and not on frequency domain? Overall SpecAugment can be applied to the raw wave (using low-pass filter for frequency domain) but there could be other reasons to apply augmentation only after feature extraction (e.g. to avoid some boundary effects as we emulate FFT in feature extractor). Maybe it is better to reformulate this aspect in the text.
- usage of seq2seq is confusing. Often it is referred to a specific architecture and loss - encoder-decoder with cross entropy, e.g. machine translation, not to a general task itself of transforming one sequence to another sequence. Worth to say definition / change notation to properly refer to sequence modeling.
- time unrolling and time merging operations: how padding can affect the training here (say I have a bunch of very short audio and very long audio in the data, even if I cut to 5s I still can have 1s audio in the batch).
- why does time unrolling ensure that all frames cannot be different? why does this not happen for time merging too as soon as time dim again is reduced? why not having the cross correlation matrix of size TxT where we reshape B and F together?
- introduce HPO notation before its usage in the text (page 4), and also be consistent with "finetuning", "fine-tuning"
- "ensures similarly size of gradients" -> it is not true as it ensures only loss scale, while gradient can be different. More careful formulation in the text is needed.
- about augmenting the target branch - maybe it is worth to reference to fixmatch paper too as this is what they are doing too.
- projection head: it is not clear if authors use it in pre-training or not (I guess not).
- why do authors crop all audio to 5 sec during pre-training? how was this value established?
- about combination contrastive and then non-contrastive training there could be combination of non-contrastive and later contrastive or even jointly too losses. Any idea why not to try it or any expectations how it will work?
- "we fine-tune models for 400k" - why so long??? in wav2vec fine-tuning is done for 20k (10h) and 80k (100h) e.g.
- beam search decoding - I would recommend from my practice is to use random search instead of grid search, next try larger language model weights as 2 could be small especially for 10h where LM contribution is huge. And also use [-10, 10] range for word insertion penalty. At least worth to check best params from wav2vec 2.0 paper and include those in the search ranges.
- General comment on improving readability: there are a lot of repetitions of the content, e.g. description of evaluation/training and then repeating the same thing in every section on results. This can be simplified.
- page 7 usage of 6/12% relative improvement: when I computed it I got way less percentage of improvement. Could authors recheck here claims and specify exactly how they got these numbers?
- Table 2 "significant difference" - looking at numbers I would say they are similar, not significant different. Also in all results we can expect at least 0.1 WER std from my experience and from many papers where people report results. So in many cases improvements which are reported are not significant and it is more "in the same ballpark" which is still valid in the sense of simplifying the SSL training and speeding it up. No need to oversell results if they give same results but with way less resources. This is already good point and justifies the need of new method.
- What do authors want to show with reporting CER? Especially after decoding?
- Batch in wav2vec 2.0 is not needed for contrastive training itself as in vision, as we sample negatives from the same sample in future/past, not from other samples. Mostly large batch is needed to efficiently train transformer architecture itself. I would suggest to reformulate a bit the sentences on this topic in the text.


**Details Of Ethics Concerns:**

No concerns. It is general SSL pre-training algorithm for speech recognition task.

**Strength And Weaknesses:**

**Strength**
- adoption of vision idea on Barlow-Twins to sequence domain
- experiments with different amount of supervision (10h, 100h, 960h) of Librispeech
- ablation on usage non-contrastive pre-training after contrastive one

**Weaknesses**
- Beam search is very shallow with not proper hyper-parameters. Yes, authors say about more practical use case, but beam search of CTC is still cheap, and it is research paper where we need to understand all limits and boundaries, especially in low-resource where contribution of language model is huge. I am ok of having several points of decoding for restricted setup and full setup. There are a lot of offline applications where we can use better / larger language models and more expensive search. With current setting it looks more artificial than practical and results obtained with language model are very-very bad, so hard to compare what exactly we are learning with SSL and how language is incorporated there. Because of transformer architecture we could learn some phoneme-level LM during SSL (and this is happening). There could be different effects of LM in the current paper in wav2vec as authors are doing 5s crop for input audio in SSL.
- "Nevertheless, trained on LibriSpeech dataset, our baselines are less than 1% absolute WER difference only when compared with SOTA solutions, e.g., Wav2Vec-2.0 (Baevski et al., 2020) which are the result of extensive HPO." - when I compare results for 10h, 100h, and 960 (Tables 1, 3, 4) of supervision with wav2vec 2.0 paper I don't see less than 1% absolute gap, it is much larger. I see a problem right now exactly with hyper-parameter search, reimplementation (or rerunning) wav2vec 2.0 baseline which is very weak and not consistent with what people reported (especially in the context that pretrained model is used and only fine-tuning is done). Also official recipe is 20k/80k fine-tuning steps only, not 400k as authors suggested to do in the paper.
- This also raises questions on the efficiency: how do authors computed numbers for training time/gpu time for wav2vec 2.0? from 400k fine-tuning? from pre-training + fine-tuning?
- I think the main weakness is reproduction of wav2vec 2.0 baseline which is far away from what is reported in prior work. Then comparison with proposed new methods looks very weird with these numbers. Next, results in Table 1, 3 for decoding with LM looks in some places very weird as not improvement is obtained even. Especially in 10h regime LM should help by a lot as we learn the phonemes / letters but very bad in word reconstruction and wav2vec 2.0 should gain huge boost with LM which is not the case authors reported. Overall results looks very weird and strange for me.
- absence of baselines on non-contrastive training, e.g. HUBERT, data2vec. (maybe at least comparison on speed in case authors can reproduce results from the paper)

Overall I feel this paper needs more work on experimental side to prove usefulness of the method. Right now it is more like proof of concept that Barlow-Twins loss can be trained with speech. No extra ablations on the parameters are given and how they are important or not (e.g. weak/strong augmentation, loss weights, EMA decay, etc.)

**Summary Of The Paper:**

Wav2vec 2.0 is widespread among speech community and used extensively to pre-train speech models for later fin-tuning on the speech recognition task. The downsides of wav2vec 2.0 is necessity of huge resources to get state-of-the-art on Librispeech, hard to tune as there are a lot of hyper-parameters and also contrastive loss which is often hard to deal. To democratize pre-training pipeline to people, authors in this paper propose to use non-contrastive loss and adapt Barlow-Twins from vision domain. They propose simple trick where reshaping is done of 3d tensor to 2d: time is stacked either with batch dim or with feature dim and then standard Barlow-Twins loss from vision is computed. These two operations, time unrolling and merging, ensure that all frames are different and that all sequence are different. With experiments on Librispeech and different amount of supervision (10h, 100h, 960) authors demonstrate that proposed method is in the same ballpark or better than wav2vec 2.0 and more GPU-efficient (can be trained with less resources, smaller batches, 2.3x faster in GPU hours). Also non-contrasitve training can be done after contrastive training and improve further results.

**Summary Of The Review:**

There are a lot of methods proposed in computer vision which can not readily be applied to other domains, e.g. sequences in speech and text. This works aims to adapt Barlow-Twins SSL technique to sequences and show that non-contrastive training can be on pair with contrastive training, e.g. wav2vec 2.0. Authors proposed to pack time either with batch dim or feature dim to compute then cross correlation matrices in Barlow-Twins. I like that there is direction developed in adapting some methods from computer vision and also developing non-contrastive methods to simplify and speed up the training. However, empirical comparison done in the paper raises a bunch of questions I specified in the main review (baselines are significantly worse than reported in original paper, beam search is very weak, etc.). Also there is no comparison with other developed speech specific non-constrastive methods like HUBERT, or more general framework as data2vec which is the most closed to considered Barlow-Twins in terms of architecture and EMA (but not loss).

---

> ### Author Response · Authors · 2022-11-12
> **Replies part-3**
>
> **"we fine-tune models for 400k" - why so long??? in wav2vec fine-tuning is done for 20k (10h) and 80k (100h) e.g.**
>
> Thanks for pointing this out. We have made a mistake with some last minute changes to the paper. The table below shows the number of steps each model was trained. 400K steps is the number used when fine-tuning with 960 hours of data, whereas the experiments with 100 and 10 hours used respectively 100K and 30K steps. Since we do not search for LR and other hyper-parameters for 10hrs and 100hrs fine-tuning settings, we used a slightly increased number of steps in the corresponding fine-tuning phases, namely 26K and 83K instead of 20K and 80K respectively.
>
> Fine-tuning Data | Steps (with early stopping) | Max Steps
> ---------------|:----------------:|:---------------:
> LL-10 | 26K | 30K
> LS-100 | 83K | 100K
> LS-960 | 359K | 400K
>
>
>
> **General comment on improving readability: there are a lot of repetitions of the content, e.g. description of evaluation/training and then repeating the same thing in every section on results. This can be simplified.**
>
> Thanks for the suggestion. We will fix this.
>
> **page 7 usage of 6/12% relative improvement: when I computed it I got way less percentage of improvement. Could authors recheck here claims and specify exactly how they got these numbers?**
>
> We say `our proposed Non-Contrastive SSL method outperforms the SOTA W2V-2 approach for speech representation learning by achieving up to 6% relative WER improvement on dev/test-clean splits of LibriSpeech dataset` which is based on following:
>
> Dev-clean: 1-(4.01/4.33) = 7.39%
>
> Test-clean: 1-(4.29/4.47) = 4.02%
>
> And we say `the Iteratively Combined SSL approach significantly boosts the performance of downstream ASR task and achieves up to 12% relative WER improvement on dev/test-clean splits`, which is based on following:
>
> Dev-clean: 1-(3.81/4.33) = 12.01%
>
> Test-clean: 1-(4.01/4.47) = 10.29%
>
>
> **Table 2 "significant difference" - looking at numbers I would say they are similar, not significant different. Also in all results we can expect at least 0.1 WER std from my experience and from many papers where people report results. So in many cases improvements which are reported are not significant and it is more "in the same ballpark" which is still valid in the sense of simplifying the SSL training and speeding it up. No need to oversell results if they give same results but with way less resources. This is already good point and justifies the need of new method.**
>
> Thanks for the suggestion. We will rephrase that sentence in the caption appropriately.
>
> **What do authors want to show with reporting CER? Especially after decoding?**
>
> We have added those results to report fine-grained analysis of our models showing the prediction accuracy (i.e., without LM) and the impact of the language model (i.e., decoding with LM). However, as a result of the response to all the reviewers, the amount of text and tables is increasing, so we’ll move the CER tables to the appendix.
>
> **Batch in wav2vec 2.0 is not needed for contrastive training itself as in vision, as we sample negatives from the same sample in future/past, not from other samples. Mostly large batch is needed to efficiently train transformer architecture itself. I would suggest to reformulate a bit the sentences on this topic in the text.**
>
> We agree that Wav2Vec-2.0 training does not require huge batches to contain negative samples. However, it requires longer sequences which makes the total batch size to 1.6hrs of audio data per batch. We will rephrase the sentence in the paper to clarify this.
>
>
> **When I compare results for 10h, 100h, and 960 (Tables 1, 3, 4) of supervision with wav2vec 2.0 paper I don't see less than 1% absolute gap, it is much larger. I see a problem right now exactly with hyper-parameter search, reimplementation (or rerunning) wav2vec 2.0 baseline which is very weak and not consistent with what people reported (especially in the context that pretrained model is used and only fine-tuning is done).**
>
> We have a maximum of 1.48% WER difference (without the use of language model). Below are the results for dev/test-clean splits.
>
> &nbsp; | 10hrs | 100hrs | 960hrs
> ---------------|:----------------:|:---------------:|:---------------:
> **Wav2Vec-2.0 from paper** | 10.9 / 11.1 | 6.1 / 6.1 | 3.2 / 3.4
> **Wav2Vec-2.0 our implementation** | 12.18 / 12.58 | 6.32 / 6.65| 4.33 / 4.47
> **Difference** |1.28 / 1.48 | 0.22 / 0.55 | 1.13 / 1.07
>
>
> **How do authors computed numbers for training time/gpu time for wav2vec 2.0? from 400k fine-tuning? from pre-training + fine-tuning?**
>
> The training time reported in Table 5 is computed for only the pre-training phase.

---

> > ### Comment · Reviewer_hCQK · 2022-11-25
> > **Discussion**
> >
> > Dear Authors,
> >
> > I went through all your comments, additional experiments and revision. I think the paper lacks two main things:
> > - absence of comparison with data2vec (or other non-contrastive) methods (Reviewer 5kcU also pointed this). Data2vec is not multi-modal. It is designed to accept any modality as input. However experiments in the paper are done on every modality separately, that is why your results should be directly comparable if you use exactly same architecture and data.
> > - Librispeech experiments are still poor. Yes, HPO on language model fixed some issues with LM decoding, but baseline without LM is still poor and is not comparable with the prior work on wav2vec 2.0. Yes you report that on clean sets it is 1% WER absolute gap (which is huge for clean set) with your reimplementation. But if you look at the other set - it is more than absolute 7% WER for 10h, almost 6% WER on 100h and 2% for 960h settings. Having in mind that you use the official released checkpoint of wav2vec 2.0 pre-training (so that you do only fine-tuning reproduction) and that I did reproduction of fine-tuning on my own before getting similar results as in the paper (+/- 0.3% WER absolute) I believe your baseline is wrongly constructed / reproduced and thus experiments are needed to be redo.
> >
> > Best regards, Reviewer.

---

> > > ### Author Response · Authors · 2022-12-07
> > > **Discussion reply**
> > >
> > > Regarding the first point, we have answered this above and copying our reply here for convenience.
> > >
> > > >Our viewpoint on this is that our work is about resource efficient training rather than decreasing WER to establish a new SOTA on LibriSpeech. In that line, we would like to clarify our viewpoint in detail as follows. From the training efficiency point of view, we note that like our paper, data2vec also employs a siamese network and learns in a non-contrastive fashion, but does so via feature masking and prediction between teacher and student rather than our proposed approach which instead brings together a Barlow–Twins like loss to the audio domain, to dramatically improve training efficiency. This is an important point for simplifying SSL training for resource efficiency as also highlighted by reviewer hCQK. Specifically, we’d like to highlight that even though data2vec decreases the required batch size from 1.6 hours (as in wav2vec2) to 1.05 hours, our approach decreases it by a much more significant margin to 0.26 hours, which in our mind speaks to the advantages of our method in terms of decreased required batch size. Also from a training time point of view, data2vec is much slower than our approach. In particular, data2vec performs the same number of steps (400K) as we do, but their batch size is much larger than ours, specifically 4x larger (1.05 / 0.26 = 4). Even though data2vec doesn’t report training time numbers. Given the fact that both data2vec and our approach employ (even if in a different manner) siamese networks but data2vec uses 4x larger batch size, indicates that data2vec is significantly slower to train than our approach.
> > >
> > >
> > > Regarding the second point, our data pipeline implementation and that of wav2vec2 is different. This may be responsible for the difference in the attained WER results, but and therefore we believe a more appropriate comparison is between our wav2vec2 results versus our non-contrastive approach versus our sequentially combined approach since these are all trained on the same data pipeline and hardware under various configurations. There we see clearly the advantages of our approach in terms of relative WER improvement. Moreover, as mentioned in the previous point the objective of this paper is not to beat SOTA in terms of WER, but instead propose a more resource efficient training paradigm for SSL ASR.

---

> > > > ### Comment · Reviewer_hCQK · 2022-12-07
> > > > **Discussion**
> > > >
> > > > Dear Authors,
> > > >
> > > >
> > > > Could you clarify at what extend and what are exact changes you did for "our data pipeline implementation and that of wav2vec 2.0 is different"? Hardware will not influence (maybe a bit, I reproduced the whole pipeline using A100, so it got me similar results) the wav2vec reproduction, also you used the same amount of steps. For now I believe you have not appropriate baselines especially in the light that you use pretrained open-sources wav2vec 2.0 model and only finetuned it. Could you also clarify what exactly changes you did in data pipeline? I agree with the point that the paper central topic is efficiency of the method due to reduction of the batch size and usage of the same number of training steps (which save resources by a lot). But in this case you should either get same or near same WER as wav2vec 2.0 or provide the trade of between amount of resources and the expected WER. For now seems you used the same resources as in wav2vec 2.0 paper to reproduce it but got way worse results than in the prior paper.
> > > >
> > > > Regarding data2vec I partially agree with your points, but data2vec achieves same/better results than wav2vec 2.0 and it could be that the large batch is a necessary part to beat wav2vec 2.0. In this light until you do fair comparison with the baselines setting the same / best conditions and showing that with the same WER you are faster in terms of resources, I cannot judge what is going on. Besides data2vec is a baseline on non-contrastive training which also (maybe) speedup training compared to wav2vec 2.0.
> > > >
> > > > Best,
> > > > Reviewer.

---

> > > > > ### Author Response · Authors · 2022-12-08
> > > > > **Reply**
> > > > >
> > > > > Here is a list of differences between our pipeline and fairseq, mostly motivated by our focus on efficiency:
> > > > >
> > > > > 1. Regarding finetuning, there are two differences:
> > > > > - Firstly, all our experiments use one state fine-tuning (where encoder and classification block are trained together) rather than two stage fine-tuning (where only the classification block is trained for the first few steps of fine-tuning, and later both encoder and classification block are jointly trained).
> > > > >
> > > > > - Secondly, rather than Adam with tri-state learning rate scheduler as in Wav2Vec-2.0 we used Adam with exponential decay.
> > > > > The reason why opted for a simpler setup is to provide a fairer comparison between the different approaches given that the Wav2Vec-2.0 scheduler was designed to improve that particular training method which contributed to decrease their reported WER.
> > > > >
> > > > >
> > > > > 2. Regarding masking, given our setup with Barlow–Twins rather than the masking based contrastive approach, we use specaugment style time-domain masking, whereas Wav2Vec-2.0 found that performing both frequency and time domain masking is best for their  masking and prediction training setup.
> > > > >
> > > > >
> > > > > 3. Regarding regularization, in our experiments we found that dropout together with data augmentation were enough for generalization, and therefore we didn’t include the option for layer drop as used in Wav2Vec-2.0. It’s possible that this may help reduce WER also in our proposed approach and sequentially combined experiments for low resource settings, but given dropout and data augmentation were enough for high resource settings, we opted to keep the same setting across all our experiments.
> > > > >
> > > > > Since the reviewer mentions that they were able to reproduce the WER results for Wav2Vec-2.0, we would like to confirm if it was with fairseq repo or their own implementation? Our experience is that the reproducibility of Wav2Vec-2.0 based on any pipeline other than fairseq is at a natural disadvantage for the Wav2Vec-2.0 setting but may benefit the development of other techniques.
> > > > >
> > > > > Regarding data2vec: we would like to emphasize one of our previous replies which indeed confirms that data2vec is faster than wav2vec2 but significantly slower than our approach. Specifically, our approach and data2vecn both use siamese networks, but data2vec uses much higher (~4x) batch size, which indicates that data2vec is significantly slower to train than our approach.

---

> > > > > > ### Comment · Reviewer_hCQK · 2022-12-09
> > > > > > **Reply**
> > > > > >
> > > > > > Dear Authors,
> > > > > >
> > > > > > I reproduced baseline in both scenarios: running their fairseq code, and reimplementing the whole pipeline in another framework and code base (here I was able entirely match the results from the paper with small std).
> > > > > >
> > > > > > The changes you made for fine-tuning I believe do not do any significant resource decrease (as the batch size is the main part), moreover fine-tuning is not so expensive as pre-training (it is a small part of the entire resources used to pre-train). You used the same number of steps to fine-tune both wav2vec 2.0 (as in the prior work) and your method, while changing other important parameters found to give very good results for wav2vec 2.0. With this, I believe your baseline is set incorrectly to compare with. Either you take wav2vec 2.0 baseline and use the same setting for your method (and show you have comparable WER but less resources) or you tune your method to get same results as in wav2vec 2.0 paper or you do curve where you change the amount of resources (by changing the batch size) used to pre-train + fine-tune wav2vec 2.0 with their recipe (at least, but it is also not ideal setting) and show the same for yours method (but again corner cases should be used from wav2vec 2.0 paper). Right now I see that you changed the baseline to use parameters not optimal for wav2vec 2.0 and which do not affect entire computation complexity in the end and claim that you have similar / better performance with less resources.
> > > > > >
> > > > > > Best,
> > > > > > Reviewer.

---

> ### Author Response · Authors · 2022-11-12
> **Replies part-2**
>
>
> **Why does time unrolling ensure that all frames cannot be different? why does this not happen for time merging too as soon as time dim again is reduced? why not having the cross correlation matrix of size TxT where we reshape B and F together?**
>
> Perhaps there is some confusion in the writing of the first question. It may be that the reviewer is referring to the passage in the text “This enables the model to enforce diversity in each sequence, such that all frames cannot be the same”. Assuming this is the case, our response would be as follows:
>
> As introduced in the text, the intuition behind time unrolling and merging was to ensure diversity in frames within each sequence and diversity within sequences of a batch, respectively. Perhaps in the paper we can clarify this by rephrasing our motivation of time unrolling and merging from a slightly different perspective, which is discussed below.
>
> “We can see that for time unrolling the cross correlation matrix based on [B ∗T, F] promotes that the i-th feature across all B & T is different from the j-th feature. On the other hand, the correlation of [T* F, B] giving rise to time merging encourages the i-th sample across all T & F is different from the j-th sample, or to put it simply the i-th sample is different from the j-th sample within the batch.”
>
> Contrasting the above two approaches with the one suggested by the reviewer (i.e., the correlation based on the matrix [B*F, T]). Whereas the time unrolling and merging approaches work around well structured entities, namely, frames and samples within a batch. However, the option that the reviewer suggests is somewhat unstructured, since [B * F, T] would encourage i-th time across all B & F to be different from the j-th time. We agree that this would indeed prevent a constant solution, but there isn’t a clear intuition supporting this option. Additionally, creating the matrices [F, F] and [B, B] is relatively cheap compared to the [T, T] matrices given than in general T >> F, B.
>
> **Introduce HPO notation before its usage in the text (page 4), and also be consistent with "finetuning", "fine-tuning”**
>
> Thanks for pointing this out. We will fix these issues.
>
> **"ensures similarly size of gradients" -> it is not true as it ensures only loss scale, while gradient can be different. More careful formulation in the text is needed.**
>
> Thank you for pointing out this miswriting on our part, which we’ll correct. The reviewer is right that this ensures losses are scaled similarly. What we meant to write is that given that a scaling of the loss propagates to the size of the gradient update, this leads to overall better gradient update behavior.
>
>
> **Why do authors crop all audio to 5 sec during pre-training? how was this value established?**
>
> Given that the proposed non-contrastive approach, leverages a siamese network which requires twice the amount of memory for weights and twice the forward passes we had to limit the sequence length. From the point of view of selecting the (batch size, sequence length) tuple we conducted time measurement experiments to find a tuple that would take advantage of the V100 GPUs we used, and found that (24, 5 * 16000) was a good trade-off. In this regard, below is a table presenting training time for alternative options.
>
>
> &nbsp; | Seq Length: 3s | Seq Length: 5s | Seq Length: 7s
> ---------------|:----------------:|:---------------:|:---------------:
> **Batch Size 24** | 704 hrs | 1064 hrs | 1483 hrs
> **Batch Size 48** | 1216 hrs | OOM | OOM | OOM
> **Batch Size  72** |1696 hrs | OOM | OOM | OOM
>
>
>
> **about combination contrastive and then non-contrastive training there could be combination of non-contrastive and later contrastive or even jointly too losses. Any idea why not to try it or any expectations how it will work?**
>
> We agree that there could be multiple ways to combine the contrastive and non-contrastive training such as C->NC (which we report on), NC->C, versus an iterative sequence like C->NC->C… or NC->C->NC…, including the exploration of optimal schedulers to switch between contrastive and non-contrastive approaches. But we leave these for further exploration as these are not in the scope of this work.
>
> In particular, our priority for this work has been threefold: Firstly, we aimed to show that non-contrastive approaches are worth exploring in the audio domain which is currently dominated by masking-based and contrastive methods. Secondly, not only are non-contrastive approaches competitive from a WER point of view, they produce similar WER at lower wall clock time for the same computational resources. Thirdly, having established the fact that there are more than one type of methods for SSL ASR, we began an exploration on potential advantages of iteratively combining them for decreased WER as compared to using only one of them.

---

> ### Author Response · Authors · 2022-11-12
> **Replies part-1**
>
> **Introduction: reference to wav2vec 2.0 in the context of absence of high quality large labeled data for many languages - it is strange usage of reference, maybe authors meant wav2vec multilingual paper then?**
>
> Thanks for pointing this out. We will change the reference to the Wav2Vec XLSR paper.
>
> **Large curated unlabeled data - we have any more appeared since which actually gave huge speed up in development for SSL and semi-sup learning, e.g. LibriLight, VoxPopuli, Common Voice.**
>
> Following the reviewer’s suggestion, we will add these references to the paper.
>
> **For SSL in speech there are missing references to data2vec (Baevski, A., et al. Data2vec: A general framework for self-supervised learning in speech, vision and language, 2022, arXiv preprint arXiv:2202.03555), wav2vec-BERT (Chung, Y.A., et al. w2v-bert: Combining contrastive learning and masked language modeling for self-supervised speech pre-training, ASRU, 2021), autoregressive/non-autoregressive predictive coding e.g. from James Glass team, etc.**
>
> Due to the page limit we had to remove some related work discussion. We will extend the introduction to discuss these studies and add the corresponding references.
>
> **Figure 1. It will be very clear if we show properly the matrices in time merging and unrolling same way as used in the text (e.g. transpose the matrix for merging) and also add dimensions to all matrices on the plot to see reshape operation right away there. This can simplify reading the image and allow clear following it.**
>
> Thanks for the suggestion. We have modified the paper in two ways. First in the text, in section 2.2 bullet point ii, we'll modify the text so rather than writing the intermediate tensor dimension [B, T*F] we’ll write only the final tensor dimension [T*F, B] to avoid confusion. Second in Figure 1, we will label each tensor with its dimensions to show these details.
>
> **Do authors do only SpecAugment on time domain and not on frequency domain? Overall SpecAugment can be applied to the raw wave (using low-pass filter for frequency domain) but there could be other reasons to apply augmentation only after feature extraction (e.g. to avoid some boundary effects as we emulate FFT in feature extractor). Maybe it is better to reformulate this aspect in the text.**
>
> We apply SpecAugment only after feature extraction. This is due to the fact that the feature extractor mimics FFT style transformation with a receptive field of 25ms of audio and a 20ms hop length, and applying masks on these features avoids boundary effects. We will add this clarification in the paper.
>
> **Usage of seq2seq is confusing. Often it is referred to a specific architecture and loss - encoder-decoder with cross entropy, e.g. machine translation, not to a general task itself of transforming one sequence to another sequence. Worth to say definition / change notation to properly refer to sequence modeling.**
>
> We will replace seq2seq with “speech-to-text” modeling.
>
> **Time unrolling and time merging operations: how padding can affect the training here (say I have a bunch of very short audio and very long audio in the data, even if I cut to 5s I still can have 1s audio in the batch).**
>
> In our experiments we first filter out sequences that are smaller than 5s and then truncate the sequences to 5s of length, so there is no padding used. However, the case of smaller sequences with some padding that the reviewer mentioned could actually have adverse effects on pre-training. For time unrolling, since that loss promotes that the i-th feature across all B & T is different from the j-th feature, the padded frames would be pushed away from zeros. Similarly, as the time merging loss promotes that i-th sample is different from the j-th sample within the batch, the sequences with high amounts of padding will again force the network to push non padded frames (such as silence) away from zero. In both cases, padding would make it harder for the network to converge depending on the percentage of padding per batch.
>
>
> **about augmenting the target branch - maybe it is worth to reference to fixmatch paper too as this is what they are doing too.**
>
> Thanks for pointing this out. We will add this reference
>
> **projection head: it is not clear if authors use it in pre-training or not (I guess not).**
>
> It is used during the pre-training phase, but not while fine-tuning. We will clarify this in the paper.

---

> ### Author Response · Authors · 2022-11-12
> **Summary of revision and replies**
>
> We thank the reviewer for their comments and suggestions. In particular,
> - We have improved the discussion of related work.
> - We have updated section 2 and figure 1 to write only the final tensor dimensions (rather than before transposing) and labeled the tensors in the figure for improved clarity.
> - We improved the discussion about loss scaling.
> - We clarified that no padding is used in our experiments and discussed the effect of padding sequences of different lengths.
> - Motivated the selection of max sequence length of 5sec in the experiments.

---

> > ### Author Response · Authors · 2022-11-16
> > **Experiments to perform additional HPO for LM decoding**
> >
> >
> > **beam search decoding - I would recommend from my practice to use random search instead of grid search, next try larger language model weights as 2 could be small especially for 10h where LM contribution is huge. And also use [-10, 10] range for word insertion penalty. At least it is worth it to check the best params from wav2vec 2.0 paper and include those in the search ranges.**
> >
> > We thank the reviewer for this suggestion. We’ve accordingly done a random search for alpha and beta as well as increased the beam width to complement the results already in the paper. Specifically, for alpha we performed uniform random search in [0, 10] and for beta uniform random search in [-10, 10], and increased the beam width from 50 to 1500, which has indeed allowed the language model to result in lower WER especially for lower resource data. In particular, the results in Table 1 for 960 hours have improved as shown in:
> >
> > Model | Fine-tuning Data | dev-clean | dev-other | test-clean | test-other | &nbsp;
> > ---------------|:----------------:|:---------------:|:---------------:|:---------------:|:---------------:|:---------------:
> > **W2V-2** | LS-960 | 3.03 | 8.88 | 3.61 | 8.92 | **OLD**
> > **Non-Contrastive** | LS-960 | 2.99 | 8.98 | 3.60 | 8.83 | **OLD**
> > **Iteratively Combined** | LS-960 | 2.83 | 8.74 | 3.55 | 8.75 | **OLD**
> > **W2V-2** | LS-960 | 2.99 | 7.97 | 3.30 | 8.10 | **NEW**
> > **Non-Contrastive** | LS-960 | 2.88 | 7.87 | 3.22 | 8.07 | **NEW**
> > **Iteratively Combined** | LS-960 | 2.83 | 7.90 | 3.20 | 8.10 | **NEW**
> >
> >
> > and those in Table 3 for 100 and 10 hours have improved as summarized below:
> >
> > Model | Fine-tuning Data | dev-clean | dev-other | test-clean | test-other | &nbsp;
> > ---------------|:----------------:|:---------------:|:---------------:|:---------------:|:---------------:|:---------------:
> > **W2V-2** | LL-10 | 10.99 | 24.49 | 11.68 | 24.95 | **OLD**
> > **Non-Contrastive** | LL-10 | 10.28 | 23.41 | 12.11 | 23.89 | **OLD**
> > **W2V-2** | LS-100 | 5.59 | 18.41 | 6.11 | 18.13 | **OLD**
> > **Non-Contrastive** | LS-100 | 4.80 | 16.38 | 5.55 | 16.16 | **OLD**
> > **W2V-2** | LL-10 | 6.65 | 17.22 | 7.04 | 17.76 | **NEW**
> > **Non-Contrastive** | LL-10 | 6.38 | 16.45 | 6.64 | 17.01 | **NEW**
> > **W2V-2** | LS-100 | 4.03 | 13.64 | 4.51 | 13.71 | **NEW**
> > **Non-Contrastive** | LS-100 | 4.04 | 13.25 | 4.45 | 13.00 | **NEW**

---

### Official Review · Reviewer_5kcU · 2022-10-26

**Confidence:** 4
**Correctness:** 2
**Technical Novelty And Significance:** 3
**Empirical Novelty And Significance:** 2
**Recommendation:** 3

**Clarity, Quality, Novelty And Reproducibility:**

The paper is generally clear and the extension of Barlow-Twins for SSL on speech is novel. Enough details are presented to be able to reproduce the work.

**Strength And Weaknesses:**

**Strengths**:
  - The paper is clear and the proposed approach is easy to follow. Performance on different subsets of the Librispeech dataset shows that non-contrastive SSL can achieve similar or even better WER than the contrastive models. The models also require smaller batch sizes resulting in fewer memory and GPU resources to train.

  - I appreciate the figure that explains clearly the difference between standard non-contrastive SSL and the proposed extension to speech. All details are presented to be able to reproduce the work.

**Questions and Weakness**:

  - Prior work: Non-contrastive methods for SSL in speech are not well discussed. There are quite a few generative/predictive models which have been investigated for SSL with speech. Some examples include MockingJay, Audio ALBERT, TERA, etc.  More exhaustive list can be found in [1]. data2vec [3] is a very close and relevant non-contrastive baseline that requires discussion.

  - I do not think that the claim "Non-contrastive SSL ASR yields a $2.3\times$ training speed up compared with contrastive SSL" is completely accurate.  It seems that the GPU hours were reported from the [2].  This may not be comparable because (a) the GPU hours can vary significantly depending on the setup. [4] reports the W2V base model training to 1764 hours when trained on 8 GPUs with gradient accumulation. It would be better to re-estimate the training times for W2V architectures on the same number of GPUs (8) used by non-contrastive models and by using the most recent w2v code for a fair comparison.  This could be done by running pre-training for a reduced number of steps.

  - While it is shown that non-autoregressive models require smaller batch size, the claim "Non-contrastive SSL requires fewer GPUs" is not accurate as gradient accumulation can be used for training using the same number of GPUs.  This also leads to better GPU hours.

- Issues with experimentation
  - For the case of iteratively combined pre-training, the paper only evaluates the setting of non-contrastive pre-training starting from a pre-trained contrastive model. This model has effectively been pre-trained for $650$K iterations. This needs to be compared against the contrastive and non-contrastive models trained for $650$K iterations to understand the performance gains/losses.

  - Confounding factors and lack of proper ablation make it hard to understand the source of the performance gains. Some examples are listed below:
    - Pre-training time is reported from the wav2vec paper which was trained on $64$ GPUs. The configuration of $8$ GPUs with gradient accumulation would have been a comparable configuration.
    - Audio is cropped to 5 seconds: This would lead to shorter sequences to the Transformer encoder than the wav2vec model. Given the quadratic complexity of the transformer, the extent of performance gains from (a) shorter sequence and (b) smaller batches is not clear. An ablation with fewer pre-training steps but isolating these factors would have made it easier to understand the impact.
    - Models are fine-tuned for $400$K steps. This is substantially more than the wav2vec models which are fine-tuned for 80K and 20K steps on LS-100 and LL-10  datasets respectively. It is not clear if the gains are only due to non-contrastive pre-training or also due to longer fine-tuning. Furthermore, this could result in slower fine-tuning for the non-contrastive model.
  - Baselines: While the paper compares against the wav2vec model, a relevant baseline would have been the data2vec [3] model which follows a non-contrastive setup as well.

**References:**

[1] Self-Supervised Speech Representation Learning: A Review. IEEE Journal of Selected Topics in Signal Processing

[2] wav2vec 2.0: A Framework for Self-Supervised Learning of Speech Representations. NeurIPS 2020

[3] data2vec: A General Framework for Self-supervised Learning in Speech, Vision and Language. ICML 2022

[4] Performance-Efficiency Trade-offs in Unsupervised Pre-training for Speech Recognition. ICASSP 2022

**Summary Of The Paper:**

This paper presents a non-contrastive pre-training method based on Barlow-Twins as an alternative to the wav2vec model which uses contrastive pre-training.  This work aims to improve the computational efficiency (memory and pre-training time) of the wav2vec model. The paper introduces two extensions to Barlow-Twins loss to account for the time dimension of an acoustic feature sequence.   Given a batch with $B$ input sequences each with time steps $T$ in embedding space of size $F$, first, time unrolling stacks all time steps from all inputs to get embeddings of shape $(B * T, F)$. The cross-correlation matrix of shape $F \times F$ is encouraged to be identity to ensure feature diversity.  Next, time merging concatenates features from all time steps to obtain embeddings of shape $(B, T * F)$. Similarly encouraging the cross-correlation of shape $B \times B$ to be identity ensures that the model does not collapse. Experiments are conducted on Librispeech dataset to evaluate the efficacy of the proposed approach.

**Summary Of The Review:**

This work introduces a non-contrastive method to improve the computational efficiency of SSL on speech sequences. I have recommended rejection primarily due to the experiments with confounding factors and lack of proper ablation as explained above. This makes it hard to understand the source and extent of performance gains.

---

> ### Author Response · Authors · 2022-11-11
> **Replies part-2**
>
>
> **While it is shown that non-autoregressive models require smaller batch size, the claim "Non-contrastive SSL requires fewer GPUs" is not accurate as gradient accumulation can be used for training using the same number of GPUs. This also leads to better GPU hours.**
>
> We agree that the claim `Non-contrastive SSL requires fewer GPUs` is not very accurate and we will rephrase it  in the caption of table 6 as following:
> `Our results show that the Non-Contrastive SSL method requires smaller batch sizes as compared to the W2V2 approach.`
>
>
>
>
> **Audio is cropped to 5 seconds: This would lead to shorter sequences to the Transformer encoder than the wav2vec model. Given the quadratic complexity of the transformer, the extent of performance gains from (a) shorter sequence and (b) smaller batches is not clear. An ablation with fewer pre-training steps but isolating these factors would have made it easier to understand the impact.**
>
> Following the reviewer’s suggestion, we performed an ablation study to understand the impact of the batch size and sequence length on training time. In particular, having as a reference the setting considered in Wav2Vec-2.0, namely batch size 6 with approx. 15s in each sample resulting in a lower amount of audio per batch of 1.4min. We expand our measurements from a single setting of batch size 24 with 5s in each sample with yields 2min of audio per batch to different configurations of batch sizes 24, 48 and 72 as well as sequence lengths of 3s, 5s and 7s as allowed by our infrastructure without increasing the gradients accumulation steps (fixed at three).
> Given that the non-contrastive approach is based on a siamese network format, with exponential moving average, there are twice the weights in memory as well as twice the number of forward passes which limits how much we can increase the batch size and the sequence lengths. Even with three step gradient accumulation (as used in all other experiments in the paper) we can see in the table below that the more demanding settings result in out of memory (OOM) on V100 GPUs having 32 GB memory. Nevertheless from the table we can see that increasing the batch size results in a higher computation time than increasing the sequence lengths. This may be due to GPU contention at higher batch sizes, or that depending on the configuration of batch size and sequence length different kernels for the different operations are being selected by the backend.
>
> &nbsp; | Seq Length: 3s | Seq Length: 5s | Seq Length: 7s
> ---------------|:----------------:|:---------------:|:---------------:
> **Batch Size 24** | 704 hrs | 1064 hrs | 1483 hrs
> **Batch Size 48** | 1216 hrs | OOM | OOM | OOM
> **Batch Size  72** |1696 hrs | OOM | OOM | OOM
>
>
>
> **Models are fine-tuned for 400K steps. This is substantially more than the wav2vec models which are fine-tuned for 80K and 20K steps on LS-100 and LL-10 datasets respectively. It is not clear if the gains are only due to non-contrastive pre-training or also due to longer fine-tuning. Furthermore, this could result in slower fine-tuning for the non-contrastive model.**
>
> Thanks for pointing this out. We have made a mistake with some last minute changes to the paper. The table below shows the number of steps each model was trained. 400K steps is the number used when fine-tuning with 960 hours of data, whereas the experiments with LS-100 and LL-10 used 100K and 30K steps respectively.
> So there hasn’t been any longer fine-tuning on the LS-960 dataset compared to Wav2Vec-2.0. However, since we do not search for LR and other hyper-parameters for LS-100 and LL-10 fine-tuning settings, we used a slightly increased number of steps in the corresponding fine-tuning phases, namely 26K and 83K instead of 20K and 80K respectively.
>
> Fine-tuning Data | Steps (with early stopping) | Max Steps
> ---------------|:----------------:|:---------------:
> LL-10 | 26K | 30K
> LS-100 | 83K | 100K
> LS-960 | 359K | 400K
>
>
>
> **While the paper compares against the wav2vec model, a relevant baseline would have been the data2vec [3] model which follows a non-contrastive setup as well.**
>
> The data2vec model training is performed under a different setting with multimodal inputs (audio, text and image). Through transferability this multimodal training leads to better performance on LibriSpeech than that achieved by Wav2Vec-2.0, which would then result in an unfair comparison between the proposed non-contrastive approach training on the single modality and the contrastive based approach training on multimodality. One possibility would be to consider training data2vec with only audio data, but then this would be equivalent to Wav2Vec-2.0 since they both share the same architecture and training settings for speech data.

---

> > ### Comment · Reviewer_5kcU · 2022-12-05
> > **Discussion**
> >
> > Dear authors,
> >
> > Thank you for taking time for providing clarifications. I went through the replies and discussion with other reviewers as well. Overall, I think the paper is not ready for publication. Below, I have the key issues that still remain:
> >
> > - Comparison to strong baseline:
> >   - As noted by reviewers hCQK and 5Dwq, the baseline has significantly worse results (specially on test/dev other) than the ones reported in wav2vec 2.0 original paper. Note that the pre-trained models are publicly available as well. The LM results reported on the 10h and 100h finetune setup are closer to the greedy decoding results reported in [2]
> >   - As pointed out by other reviewers, data2vec[3] model is modality agnostic, however, it is still trained separately on each modality. The data2vec on audio results in [3] refer to model trained only on audio. This is an important baseline that needs to be compared against.
> >   - Pre-training time on 8 GPUs with gradient accumulation: In past, I have verified that simply setting the gradient accumulation factor to 8 on a 8 GPU setup gives training times similar to reported in [4]. Given this and the WER difference, I would encourage the authors to take a closer look at their wav2vec implementation.
> >
> > [2] wav2vec 2.0: A Framework for Self-Supervised Learning of Speech Representations. NeurIPS 2020
> >
> > [3] data2vec: A General Framework for Self-supervised Learning in Speech, Vision and Language. ICML 2022
> >
> > [4] Performance-Efficiency Trade-offs in Unsupervised Pre-training for Speech Recognition. ICASSP 2022

---

> > > ### Author Response · Authors · 2022-12-07
> > > **Discussion reply**
> > >
> > > We thank the reviewers for all their comments to improve our paper. It seems the reviewers find it important to compare our work against data2vec (in addition to wav2vec2), in terms of WER.
> > >
> > > Our viewpoint on this is that our work is about resource efficient training rather than decreasing WER to establish a new SOTA on LibriSpeech. In that line, we would like to clarify our viewpoint in detail as follows. From the training efficiency point of view, we note that like our paper, data2vec also employs a siamese network and learns in a non-contrastive fashion, but does so via feature masking and prediction between teacher and student rather than our proposed approach which instead brings together a Barlow–Twins like loss to the audio domain, to dramatically improve training efficiency. This is an important point for simplifying SSL training for resource efficiency as also highlighted by reviewer hCQK. Specifically, we’d like to highlight that even though data2vec decreases the required batch size from 1.6 hours (as in wav2vec2) to 1.05 hours, our approach decreases it by a much more significant margin to 0.26 hours, which in our mind speaks to the advantages of our method in terms of decreased required batch size. Also from a training time point of view, data2vec is much slower than our approach. In particular, data2vec performs the same number of steps (400K) as we do, but their batch size is much larger than ours, specifically 4x larger (1.05 / 0.26 = 4). Even though data2vec doesn’t report training time numbers. Given the fact that both data2vec and our approach employ (even if in a different manner) siamese networks but data2vec uses 4x larger batch size, indicates that data2vec is significantly slower to train than our approach.
> > >
> > >
> > > Regarding pre-training with gradient accumulation of 8 steps: We have computed the training time by following the exact same training scheme with 8 GPUs on fairseq repo. Our results show that pre-training takes 2376 GPU hours. However, the reviewer claims that it takes much less GPU hours with 8 GPUs instead of 64 GPUs. We have computed the training time based on 10K steps. Perhaps the reviewer averaged only a few iterations rather than those comprising an entire epoch, or is working from an earlier version of latest master or a different branch or fork, or working under a more high performing file system, cpu or gpu specs.

---

> ### Author Response · Authors · 2022-11-11
> **Replies part-1**
>
>
> **Discussion about comparison with prior work**
>
> We will add the following discussion about the prior works as suggested by the reviewer.
>
> ```
> Self-supervised learning for ASR by utilizing unlabeled speech data has gathered increasing interests recently. Past studies have proposed variations around the idea of learning speech representations by predicting the content of the unseen regions [1,2,3,4]. On the other hand, recent studies have shown the potential of SSL pre-training by contrasting the target unseen frame with randomly sampled ones[5]. Similarly, studies in [5,6] have proposed to predict discrete targets of masked regions as the SSL training objective. As shown in [6] Wav2Vec-2.0 and HuBERT approaches have demonstrated a great performance by outperforming all other reconstruction or contrastive alternative approaches. However, we did not add HuBERT as a baseline in our experiments since Wav2Vec-2.0 BASE model’s performance (2.7 / 7.9, 3.4 / 8.0 on dev-clean/other, test-clean/other) is very similar to HuBERT BASE model’s performance (2.7 / 7.8, 3.4 / 8.1 on dev-clean/other, test-clean/other).
>
> [1] Ling et al. Deep contextualized acoustic representations for semi-supervised speech recognition. ICASSP 2020.
> [2] Ling et al. Decoar 2.0: Deep contextualized acoustic representations with vector quantization. arXiv 2020.
> [3] Liu et al. Mockingjay: Unsupervised speech representation learning with deep bidirectional transformer encoders. In ICASSP 2020.
> [4] Chi et al. Audio albert: A lite bert for self-supervised learning of audio representation. arXiv 2020.
> [5] Baevski et al. Wav2Vec 2.0: A framework for self-supervised learning of speech representations. NeurIPS 2020.
> [6] Hsu et al. A. Hubert: Self-supervised speech representation learning by masked prediction of hidden units. TASLP. 2021.
> [7] Baevski et al. Effectiveness of self-supervised pre-training for speech recognition. arXiv 2019.
> ```
>
> **Re-estimate the training times for W2V architectures on 8 GPUs**
>
> Following the reviewer’s suggestion, we considered the case of 8 GPUs rather than 64, with gradient accumulation of 8 steps to ensure the same amount of data is being processed per gradient update step. Given the increased number of gradient accumulation steps the training time is expected to increase, but we note that for 8 GPUs it’s possible to allocate all the V100’s in one node, thereby decreasing communication cost significantly. The comparison of the two settings is presented in the table below. We will add these to the paper and modify the 2.3x to 2.23x gain in training time.
>
> Model | GPU Hours | No. of GPUs | Wall Clock Hours
> ---------------|:----------------:|:---------------:|:---------------:
> Wav2Vec-2.0 | 2457| 64 | 38
> Wav2Vec-2.0 | 2376| 8 | 297
> Non-Contrastive | 1064 | 8 | 133
>
>
> The reference [4] that the reviewer points out mentions `Notably, W2V2-base is reported as taking 102.4 GPU-days to pre-train on 64 GPUs, but from our estimation it only takes 73.5 GPU-days on 8 GPUs.` (73.5 days = 1764 hours) and `We use 8 gradient accumulation steps to simulate 64-GPU training with 8 GPUs`, however the paper doesn’t show how exactly their estimation was performed. Given this, we took fairseq’s repo and set up the same experiment with 8 NVIDIA V100 GPUs and gradient accumulation steps to 8, which gave higher training time as shown above in the table (2nd row). It’s possible that the increased training time is due to poorer disk access or CPU performance available to us in our setup in a shared system, which is the one we use in all our experiments.
>
>
> **For the case of iteratively combined pre-training, the paper only evaluates the setting of non-contrastive pre-training starting from a pre-trained contrastive model. This model has effectively been pre-trained for 650K iterations. This needs to be compared against the contrastive and non-contrastive models trained for 650K iterations.**
>
> Our priority for this work has been threefold: Firstly, we aimed to show that non-contrastive approaches are worth exploring in the audio domain, which is currently dominated by masking-based and contrastive methods. Secondly, not only are non-contrastive approaches competitive from a WER point of view, they produce similar WER at lower wall clock time for the same computational resources. Thirdly, having established the fact that there are more than one type of methods for SSL ASR, we began an exploration on the potential advantages of iteratively combining them for decreased WER as compared to using only one of them. We believe the experimental scope of this work has served to establish the first two points, but agree the third point requires further exploration. In particular, C->NC (which we report on), NC->C, versus an iterative sequence such as C->NC->C… or NC->C->NC…, including the exploration of optimal schedulers to switch between contrastive and non-contrastive approaches, which we leave for further exploration.

---

> ### Author Response · Authors · 2022-11-11
> **Summary of revision and replies**
>
> First of all, we thank the reviewer for their time and constructive feedback. In particular:
> - We have expanded the discussion of related work.
> - Updated Table 5 so that each method being compared has been trained on the same number of GPUs.
> - Expanded on future work, in particular related to different schedulers for combining proposed non-contrastive and contrastive approaches.
> - Added an ablation study to understand the impact of the batch size and sequence length on training time.
> - Confirmed the number of fine-tuning steps for 960 hours, and corrected the number of fine-tuning steps in 10 and 100 hours experiments.

---

### Decision · Program_Chairs · 2023-01-20

**Decision:**

Reject

**Justification For Why Not Higher Score:**

* better experimental rigor and explanations

**Justification For Why Not Lower Score:**

* interesting idea

**Metareview: Summary, Strengths And Weaknesses:**

This paper looks at an alternative to wav2vec by discussing a non-constrastive pre-training method

Strengths:
* well written and clear
* novelty with Barlow-Twins to speech domain
Weaknesses
* prior work is lacking
* issues with some of the claims
* experiments are lacking